# Membrane cholesterol regulates inhibition and substrate transport by the glycine transporter, GlyT2

Zachary J Frangos[1,*], Katie A Wilson[2,3,*] , Heather M Aitken[2,4], Ryan Cantwell Chater[1] , Robert J Vandenberg[1] , Megan L O'Mara[2,4]

**Membrane cholesterol binds to and modulates the function of various SLC6 neurotransmitter transporters, including stabilizing the outward-facing conformation of the dopamine and serotonin transporters. Here, we investigate how cholesterol binds to GlyT2 (SLC6A5), modulates glycine transport rate, and influences bioactive lipid inhibition of GlyT2. Bioactive lipid inhibitors are analgesics that bind to an allosteric site accessible from the extracellular solution when GlyT2 adopts an outward-facing conformation. Using molecular dynamics simulations, mutagenesis, and cholesterol depletion experiments, we show that bioactive lipid inhibition of glycine transport is modulated by the recruitment of membrane cholesterol to a binding site formed by transmembrane helices 1, 5, and 7. Recruitment involves cholesterol flipping from its membrane orientation, and insertion of the 3' hydroxyl group into the cholesterol binding cavity, close to the allosteric site. The synergy between cholesterol and allosteric inhibitors provides a novel mechanism of inhibition and a potential avenue for the development of potent GlyT2 inhibitors as alternative therapeutics for the treatment of neuropathic pain and therapeutics that target other SLC6 transporters.**

## Introduction

The glycine transporter 2 (GlyT2) is a member of the SLC6 family of neurotransmitter transporters that plays an important role in regulating glycine concentrations in inhibitory synapses of the ascending pain pathway. After the presynaptic release of glycine, GlyT2 transports glycine from the synaptic cleft and back into the presynaptic neuron where it can be recycled into synaptic vesicles to maintain glycinergic neurotransmission. SLC6 transporters are secondary active transporters that exploit $Na^+$ gradients to drive the transport of substrate across cell membranes through a conserved general mechanism (Kristensen et al, 2011). Substrate and ions bind to an outward-open conformation of the transporter. The transporter then undergoes a series of conformational changes resulting in an inward-open conformation that allows the release of substrate to the intracellular environment.

GlyT2 and other SLC6 transporters localize in cholesterol-enriched lipid raft domains in the plasma membrane (Scanlon et al, 2001; Magnani et al, 2004; Foster et al, 2008; Núñez et al, 2008; Liu et al, 2009; Jones et al, 2012), and their activity is modulated by membrane cholesterol concentration (Scanlon et al, 2001). Recent crystal and cryoEM structures, and molecular dynamics (MD) simulations, have identified five cholesterol binding sites (CHOL1-5) in homologous SLC6 transporters, namely, the drosophila dopamine transporter (dDAT), human dopamine transporter, and human serotonin transporter (SERT) (Penmatsa et al, 2013; Wang et al, 2015; Coleman et al, 2016; Ferraro et al, 2016; Zeppelin et al, 2018). Depletion of membrane cholesterol by methyl-$\beta$-cyclodextrin (M$\beta$CD) reduces transporter activity (Scanlon et al, 2001; Magnani et al, 2004; Foster et al, 2008; Núñez et al, 2008; Liu et al, 2009; Jones et al, 2012), whereas experimental and MD work suggests that the binding of cholesterol changes the transporter conformational equilibrium to favour the outward-facing conformation (Hong & Amara, 2010; Bjerregaard et al, 2015; Laursen et al, 2018; Zeppelin et al, 2018). Further coarse-grained MD studies comparing the lipid annulus of select SLC6 transporters show transporter-specific differences in cholesterol occupancies across these crystallographic binding sites (Wilson et al, 2021b).

In dDAT, CHOL1 is formed by TM1a, TM5, and TM7, which also forms part of the transporter's core domain that undergoes significant conformational changes during the transition from the outward-facing to the inward-facing state of the transporter (Zeppelin et al, 2018). Thus, cholesterol bound to the CHOL1 site is most likely to be responsible for the regulatory actions of cholesterol on transporter function. This is supported by MD simulations of human dopamine transporter, where cholesterol bound to CHOL1 prevents conformational changes that mediate intracellular gating such as unwinding of TM5, an essential component in the early stages of the

[1]Molecular Biomedicine Theme, School of Medical Sciences, University of Sydney, Sydney, Australia  [2]Research School of Chemistry, College of Science, The Australian National University, Canberra, Australia  [3]Department of Biochemistry, Memorial University of Newfoundland, St. John's, Canada  [4]Australian Institute of Bioengineering and Nanotechnology, The University of Queensland, St Lucia, Australia

Correspondence: m.omara@uq.edu.au
*Zachary J Frangos and Katie A Wilson are co-first authors

transition to the inward-facing state (Stolzenberg et al, 2017; Zeppelin et al, 2018). Furthermore, treatment with MβCD and introduction of polar residues into CHOL1 of hSERT through mutagenesis disrupts cholesterol binding and results in a favouring of the inward-facing conformation (Bjerregaard et al, 2015; Laursen et al, 2018). This has led to the notion that cholesterol bound to the CHOL1 site stabilizes the outward-facing state of neurotransmitter sodium symporters (NSSs) and forms a dynamic part of the transport cycle, regulating transitions between the outward and inward conformations.

Membrane cholesterol content has also been shown to alter the affinity of NSS ligands, based on the conformational state they preferentially bind. Depleting membrane cholesterol shifts the conformational equilibrium of transporters towards the inward-facing state and enhances the activity of compounds that bind to this conformation and decreases the activity of ligands that bind the outward-facing conformation (Scanlon et al, 2001; Laursen et al, 2018). Similarly, supplementing the membrane with cholesterol shifts the conformational equilibrium towards the outward-facing state and increases the activity of compounds that preferentially bind this conformation (Hong & Amara, 2010).

Inhibitors of GlyT2 slow the removal of glycine from the synaptic cleft, prolonging glycinergic transmission to inhibit pain (Vandenberg et al, 2014, 2016). We have previously shown that N-*acyl* amino acids are a potent class of lipid-based GlyT2 inhibitors (Mostyn et al, 2017, 2019a). These inhibitors bind to a site (referred to as the lipid allosteric site, LAS) that is accessible from the extracellular solution in the outward-facing conformation (Mostyn et al, 2019b). In subsequent work, we have shown that the activity of these lipid inhibitors is influenced by the depth of penetration of the lipid tail into the LAS (Wilson et al, 2021a). Nevertheless, the interplay between lipid inhibitor and cholesterol binding to GlyT2 is unknown. In the present work, we use a combination of MD simulations, cholesterol depletion techniques, and mutagenesis to examine the interactions between bioactive lipid inhibitors bound to GlyT2 and how membrane cholesterol influences these interactions. We demonstrate that glycine transport by GlyT2 and its sensitivity to bioactive lipid inhibition are influenced by interactions with cholesterol molecules recruited from the membrane to a specific site on GlyT2.

# Results

### The LAS is only accessible in the outward-facing conformation of GlyT2

To better understand whether lipid inhibitor binding is specific to a single conformation of GlyT2, we compared the structure of the LAS in the inward- and outward-facing conformations of GlyT2. We used our previously characterized model of the outward-facing conformation (Subramanian et al, 2016; Mostyn et al, 2019b; Wilson et al, 2021a) and developed a homology model of inward-facing GlyT2, using the inward-facing cryoEM structure of human SERT as a structural template (PDB ID: 6DZZ) (Coleman et al, 2019). Triplicate 500 ns united atom simulations were performed on both the inward- and the outward-facing GlyT2 models, and the LAS pocket volume was calculated for both conformations. Consistent with

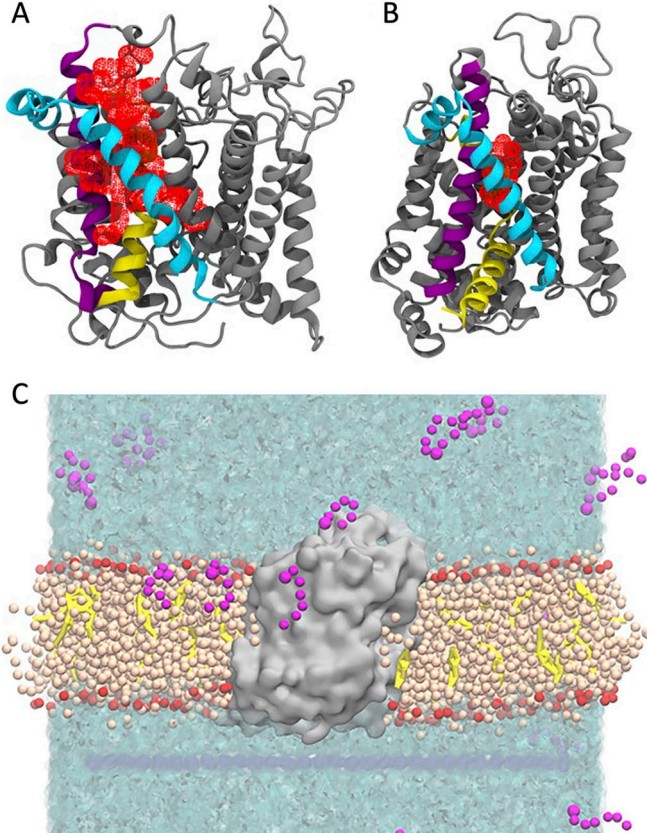

**Figure 1. LAS pocket volume for GlyT2 in the outward and inward conformations.**
**(A, B)** Pocket volume (red mesh) for the LAS in the (A) outward-open and (B) inward-open conformations of GlyT2. Important transmembrane domains are shown including TM1 (yellow), TM5 (cyan), and TM7 (purple). **(C)** Coarse-grained molecular dynamics simulation set-up showing GlyT2 (grey) embedded in an 80% POPC (tan)/20% cholesterol (yellow) membrane, with 20 molecules of a given species of lipid inhibitor (magenta) in the aqueous solution (cyan).

previous studies (Mostyn et al, 2019b; Wilson et al, 2021a), throughout all replicate simulations of outward-facing GlyT2 simulations, the LAS is accessible from the extracellular solution. The volume of the LAS cavity is between 750 and 1750 $\text{Å}^3$ for 60% of the total simulation time, and the minimum volume always exceeds 500 $\text{Å}^3$ (Fig 1A). In contrast, the LAS volume in the inward conformation is substantially smaller. For 77% of the total 1,500 ns, the LAS volume is <750 $\text{Å}^3$ (Fig 1B). This includes an extended period (510 ns or 34% of the total simulation time) in which the LAS is completely absent. Notably, this reduced volume LAS is not accessible from the extracellular solution. The observed pocket volumes indicates that a functional LAS capable of inhibitor binding is only formed in the GlyT2 outward-facing conformation.

### Lipid inhibitors interact with membrane cholesterol

To investigate the spontaneous binding of lipid inhibitors to the GlyT2 LAS, coarse-grained (CG) MD simulations were performed on the GlyT2 outward-facing conformation embedded in an 80% POPC/20% cholesterol membrane, with 20 molecules of a given

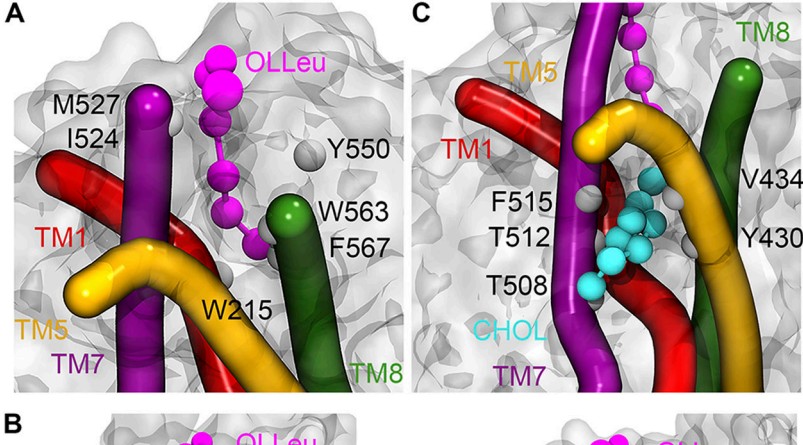

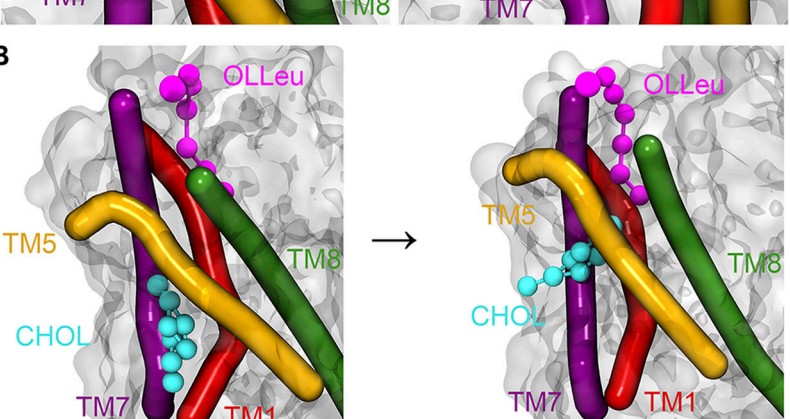

**Figure 2. Representative snapshot from CG spontaneous binding simulations showing OLLeu and associated CHOL bound in the LAS of GlyT2.**
**(A)** OLLeu bound in the LAS with residues that are in contact with the lipid inhibitor for >60% of the total simulation time highlighted. **(B)** CHOL initial bound in the dDAT CHOL1 site before flipping orientations to burrow between TM5 and TM7 at the bottom of the LAS. **(C)** CHOL bound at the bottom of the LAS site with residues that are in contact with the lipid inhibitor for >75% of the simulation time that CHOL is interacting with GlyT2 highlighted. Important transmembrane domains are shown including TM1 (red), TM5 (yellow), TM7 (purple), and TM8 (green) with cholesterol (cyan) and OLLeu (magenta) presented as spheres.

species of lipid inhibitor in the aqueous solution, as shown in Fig 1C. Four different lipid inhibitors were examined in these spontaneous binding simulations, namely, oleoyl-L-tryptophan (OLTrp), oleoyl-L-serine (OLSer), oleoyl-L-leucine (OLLeu), and oleoyl-L-lysine (OLLys). Most of the lipid inhibitors adsorbed to the surface of the membrane or formed micelles before adsorbing and partitioning into the membrane (Fig S1). The partitioning of the lipids into the membrane did not significantly alter overall membrane properties (Schumann-Gillett & O'Mara, 2019; Wilson et al, 2021a) (Table S1). In one replicate simulation, a single OLLeu lipid inhibitor entered the LAS from the extracellular solution after 0.05 μs of simulation, where it remained bound for a further 9.3 μs, before dissociating from the LAS and entering the membrane. While bound in the LAS, OLLeu interacted with residues from EL4, TM7, and TM8 (Fig 2A and Table S2). In particular, OLLeu interacts with F567, I524, M527, and W563 for >8 μs and with W215 and Y550 for >6 μs during the 10-μs simulation. The combined effect of these interactions is likely responsible for stabilizing OLLeu in this site. Notably, Y550 and W563 have been previously shown through mutagenesis experiments to affect the activity of lipid inhibitors on GlyT2 (Mostyn et al, 2019b). When considered together, this demonstrates that a lipid inhibitor can spontaneously bind from solution to the GlyT2 LAS and that the binding pose adopted is consistent with that observed in previous docking and atomistic simulations (Mostyn et al, 2019b; Wilson et al, 2021a).

While OLLeu was bound in the LAS, a previously unseen interaction occurs between OLLeu and membrane cholesterol. After 1 μs of CG simulation, a molecule of cholesterol within the membrane associates with TM1a, TM5, and TM7 in a similar orientation to that observed for cholesterol bound at the CHOL1 site of dDAT. However, after 2.6 μs, the cholesterol molecule flips to adopt the opposite orientation in the membrane, burying deeper into a cavity between TM1, TM5, and TM7 of GlyT2, such that the 3′ hydroxyl group of cholesterol forms a close association with the terminal end of the bioactive lipid tail (Fig 2B) and the isooctyl tail of cholesterol is partitioned into the phospholipid acyl tails and oriented at a near-perpendicular angle to the plane of the membrane. Cholesterol remains bound to the cavity for 5 μs of the CG simulation. In this flipped orientation, the cholesterol molecule is in direct contact with the bound OLLeu for 30% of the time cholesterol is bound. Upon entering the cholesterol binding cavity formed by TM1, TM5, and TM7, the cholesterol molecule is within 6.0 Å of Y430, L433, V434, T508, A511, T512, and F515 for >75% of the time it is bound (Fig 2C and Table S3). The combined effect of these interactions is likely responsible for stabilizing cholesterol in this site. After 5 μs of CG simulation, cholesterol dissociates from the binding cavity and reorients within the bulk of the membrane. Without cholesterol bound between TM1a, TM5, and TM7, OLLeu dissociates, suggesting that the interaction between OLLeu and membrane cholesterol may be an important factor in the binding and mechanism of inhibition of these lipids.

To further understand how the function of GlyT2 is modulated by the binding of the lipid inhibitor to the extracellular allosteric pocket, a representative frame from the CG simulation containing both bound cholesterol and OLLeu was backmapped to atomistic

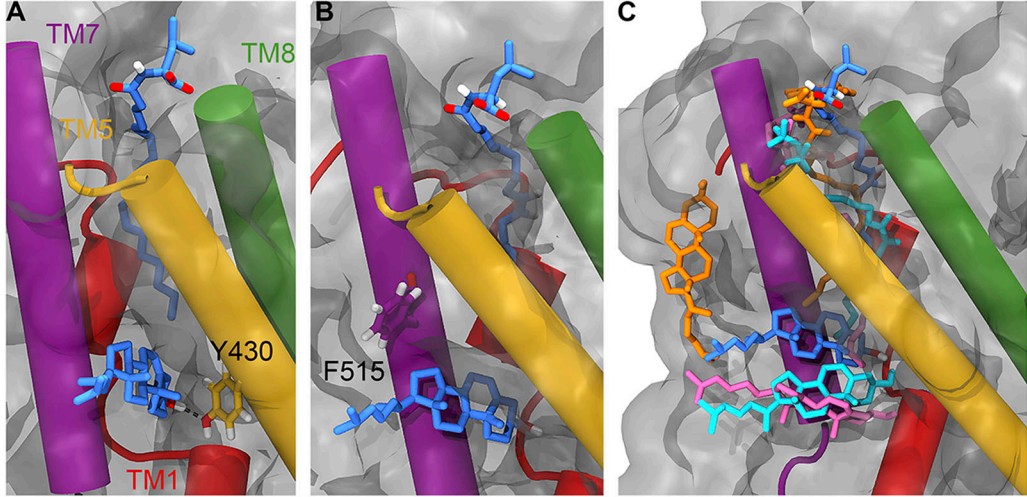

**Figure 3. Position of the lipid inhibitor and associated CHOL after 500 ns of atomistic MD simulations.**
**(A)** Position of Y430 (yellow liquorice) relative to OLLeu (blue liquorice, upper) and bound cholesterol (blue liquorice, lower). **(B)** Position of F515 (purple liquorice) relative to OLLeu (blue liquorice, upper) and bound cholesterol (blue liquorice, lower). **(C)** Overlay of the relative position of OLLeu (blue), OLLys (pink), OLCarn (cyan), and OLTrp (orange) and cholesterol in the LAS from 500-ns simulations. Cholesterol is shown in the same colour as the lipid inhibitor from the corresponding simulation. Important transmembrane domains are shown including TM1 (red), TM5 (yellow), TM7 (purple), and TM8 (green).

detail, and the system simulated in triplicate for a further 500 ns using the GROMOS 54a7 forcefield (Schmid et al, 2011). In all three replicates, the lipid inhibitor remained in the LAS located between EL4, TM5, TM7, and TM8 (Fig 3). Although there were variations in the precise orientation of the lipid inhibitor tail across the three replicate simulations, the lipid inhibitor formed direct contacts (within 4 Å) with P218, L437, I520, V523, M527, F562, W563, I566, F567, and M570 for >50% of the combined 1,500 ns of the backmapped atomistic simulations (Table S4). This is consistent with the contact residues observed in the CG simulations and those previously proposed in a combined mutagenesis/docking study (Mostyn et al, 2019b).

The atomistic simulations provide a deeper biochemical insight into how the bound cholesterol is interacting with GlyT2 and the lipid inhibitor. Although cholesterol remains bound in the LAS for ~1,050 ns of the combined 1,500 ns of simulation time, the precise interactions formed by the bound cholesterol molecule are dependent on the relative orientation of the OLLeu lipid inhibitor in each of the three replicate simulations. When the inhibitor tail is extended, the side chain of Y430, located at the base of the extracellular allosteric pocket, is oriented into the cavity between TM1a, TM5, and TM7 (Fig 3). In this orientation, cholesterol binding is primarily stabilized through its interaction with Y430, through both hydrogen bonding (with the 3′ hydroxyl) and π-stacking interactions. Without this reorientation of Y430, cholesterol dissociates from the LAS. Interactions between Y430 and cholesterol occur for 65% (975 ns) of the combined 1,500-ns simulation. Cholesterol binding is further stabilized through a dynamic hydrogen bonding network with residues within the TM1a, TM5, and TM7 cavity, mediated by V205 and through transient contacts with the nearby residues T427 and T573. Residues C507, T508, A511, T512, and F515 are also in close proximity to bound cholesterol molecules (Table S5). Overall, these atomistic simulations suggest that the position of the lipid tail and reorientation of Y430 (Fig 3A) dictate the binding of cholesterol.

As the spontaneous binding of a single lipid inhibitor to the LAS of GlyT2 is a statistically rare event in a simulation system containing 20 inhibitors, the interactions between cholesterol and three different lipid inhibitors, OLLys, oleoyl-L-carnitine (OLCarn), and OLTrp, were investigated by docking the inhibitors to the LAS in the backmapped GlyT2 system with bound cholesterol. Consistent with the observations for the OLLeu system, in the presence of OLLys or OLCarn, cholesterol remains near the terminal end of the lipid inhibitor tail, in the cavity between TM1a, TM5, and TM7 for >1,000 ns of the combined atomistic simulation time for each lipid inhibitor (Fig 3 and Table S4). It is noteworthy that the bound conformation of cholesterol depends on the species of inhibitor bound at the LAS. Similar to OLLeu, when OLCarn or OLLys is bound to the LAS, the cholesterol molecule lies nearly perpendicular to the membrane and interacts with V205, Y430, and F515 for >50% of the total simulation time (Table S5). In this position, the cholesterol molecule forms hydrogen bonding interactions with V205, Y399, and T481. In contrast, with OLTrp bound at the LAS, cholesterol does not maintain this perpendicular orientation and is bound for less than half of the total simulation time. Here, cholesterol either rapidly dissociates from the cavity and associates with the surface of TM7, or loosely associates with the base of the LAS (Fig 3C). The most notable difference between the binding of the cholesterol-interacting bioactive lipids and OLTrp is the orientation of Y430 (TM5). The tails of OLLys, OLLeu, and OLCarn penetrate deeper into the binding cavity. This deeper inhibitor penetration is associated with the reorientation of the Y430 side chain into the cavity between TM1a, TM5, and TM7. Together, these facilitate the cholesterol hydrogen bonding and stacking interactions that stabilize its binding. In contrast, the tail of OLTrp does not penetrate as deep into the binding site and the side chain orientation of Y430 is unperturbed, preventing formation of the cholesterol binding site. Previous studies have shown the deep penetration of bioactive lipids into the LAS is critical for potent inhibition of GlyT2 (Wilson et al, 2021a).

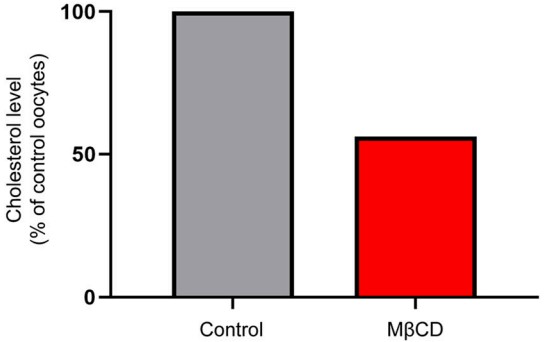

**Figure 4. Cholesterol levels of control and MβCD-treated *Xenopus laevis* oocytes.**
Groups of six oocytes were incubated in 0 mM (control) or 15 mM MβCD for 30 min at 32°C. After incubation, oocytes were lysed in 1% Triton X-100 by vortexing and manually disrupting the membranes with forceps. Cell lysates were centrifuged, and the cholesterol content of the supernatant was quantified using the Amplex Red method. Cholesterol levels were normalized to the control cells, and the data are representative of one batch of oocytes.

## MβCD treatment alters the transport activity of GlyT2 and the potency and reversibility of some GlyT2 inhibitors

To explore whether the interaction between membrane cholesterol and bioactive lipid inhibitors observed in the MD simulations influences glycine transport and lipid inhibition of GlyT2, the effects of cholesterol depletion from oocytes expressing GlyT2 were investigated. MβCD is able to deplete cholesterol from cell membranes and has been used in a number of studies to investigate the roles of cholesterol on membrane protein function (Scanlon et al, 2001; Magnani et al, 2004; Foster et al, 2008; Núñez et al, 2008; Liu et al, 2009; Hong & Amara, 2010; Jones et al, 2012; Laursen et al, 2018).

A wide range of conditions have been used to deplete cholesterol. In our study, we sought a method that can rapidly and selectively deplete cholesterol while keeping the cell viable so that we could measure the functional properties of each oocyte expressing GlyT2 before and after cholesterol depletion. Using a single oocyte for both measurements provides a highly consistent and direct measure of the effects of cholesterol. Under voltage-clamp conditions, concentration-dependent glycine transport currents were first recorded on an oocyte expressing GlyT2. The cell was then unclamped and removed from the electrophysiology rig and placed in an incubator at 32°C in a solution containing 15 mM MβCD for 30 min. The oocyte was then replaced in the electrophysiology rig and washed for 10 min in recording buffer, the voltage clamp was re-established, and the glycine transport currents were remeasured. Any cells where there was a significant change in the resting membrane potential after MβCD treatment were discarded because they may reflect damage to the cell membrane. The effectiveness of MβCD to deplete cholesterol from the oocyte membrane was estimated by measuring the cholesterol content with and without MβCD treatment. Under these conditions, MβCD caused a 44% reduction in membrane cholesterol content, which is consistent with previous studies in oocytes (Fig 4) (Santiago et al, 2001; Sadler & Jacobs, 2004).

Although enrichment of cholesterol in *Xenopus laevis* oocyte membranes using a MβCD–cholesterol complex has been found to be of low reproducibility and efficiency (Slayden et al, 2020), we attempted to restore membrane cholesterol via this method. However, given the time frame of the experiments it was difficult to distinguish between contributions of cholesterol resupplemented to the membrane and changes in transporter expression to changes in activity. Thus, to convince ourselves that the MβCD treatment predominantly depleted cholesterol from the membrane we also used a derivative of MβCD, γ-cyclodextrin (γCD). γCD will also chelate various lipids but is less effective at chelating cholesterol than MβCD (Ravichandran & Divakar, 1998).

Treatment of oocytes expressing GlyT2 with MβCD significantly reduced the maximal relative transport velocity ($V_{max}$) from 1.00 to 0.59 ($P < 0.001$; Fig 5 and Table S6). A trend towards higher apparent glycine affinity was also observed; however, this did not reach statistical significance ($P = 0.064$). These alterations in glycine transport align with previous examinations of the effect of MβCD-induced cholesterol depletion on the transport activity of GlyT2, and DAT and SERT (Scanlon et al, 2001; Núñez et al, 2008; Jones et al, 2012). Treatment of oocytes expressing GlyT2 with γCD produced a substantially smaller reduction in $V_{max}$ from 1.00 to 0.86 ($P = 0.007$; Fig 5 and Table S6). As γCD is a less effective cholesterol-sequestering agent, this result suggests that the effect of MβCD treatment is consistent with a majority cholesterol effect.

Inhibitor concentration–responses were measured to examine the effect of cholesterol depletion on the activity of the bioactive lipids studied in the MD simulations. Cholesterol depletion does not alter the potency of OLCarn or OLTrp (Fig 6 and Table S7), but significantly reduces the potency of OLLys and OLLeu by 2.3 ($P = 0.019$) and 2.4-fold ($P = 0.008$), respectively (Fig 6 and Table S7). No differences were observed in the amount of inhibition produced by application of 3 $\mu$M of any of the inhibitors (Table S7), suggesting cholesterol depletion influences the potency but not the maximal level of inhibition by the bioactive lipids. These data are consistent with MD simulations identifying interactions between cholesterol and the bound OLLys or OLLeu, and the lack of interaction between cholesterol and bound OLTrp. Taken together, these results suggest that the presence of cholesterol enhances the potency of bioactive lipids in a headgroup-dependent manner.

The MD simulations suggested that cholesterol binding to GlyT2 influences the dissociation of the lipid inhibitors from the LAS. To test this prediction, the influence of membrane cholesterol on the reversibility of GlyT2 inhibition by bioactive lipids was examined using 30-min inhibitor washout assays using the $IC_{50}$ of the inhibitor with and without cholesterol depletion. The rate of washout of OLTrp, which is not predicted to interact with cholesterol, was unchanged and was not reversed (Fig 7 and Table S8). The transport current recovery half-life after OLLeu application was unchanged, but there was a significant increase in the level of transport current recovery after 30 min from 49.4% to 90.7% ($P < 0.0001$) after cholesterol depletion. Transport current recovery half-lives were unable to be determined for OLCarn and OLLys because recovery from these compounds did not plateau within the time frame of the assay. Therefore, the level of transport current recovery after 30 min was used as a measure of reversibility. Cholesterol depletion significantly increased the transport current recovery 30 min after OLCarn treatment from 45.7%– to 92.4% ($P < 0.0001$) (Fig 7 and Table S8). The similarity in shape of the recovery curves after OLCarn inhibition indicates both conditions are likely to reach the same

**Figure 5. Membrane cholesterol depletion alters the functionality of WT GlyT2 expressed in *Xenopus laevis* oocytes.**
**(A, B)** Baseline (black curves) glycine-dependent transport currents were measured in the presence of increasing glycine concentrations (1–300 µM). Individual glycine concentrations were continuously applied until a stable current was achieved, after which point glycine was completely removed by washing in recording buffer before proceeding to the next concentration. **(A, B)** After baseline recordings, oocytes were removed from the electrophysiology rig and incubated in 0 mM (blue curve) or 15 mM (red curves) (A) methyl-$\beta$-cyclodextrin (M$\beta$CD) or (B) $\gamma$-cyclodextrin ($\gamma$CD) for 30 min at 32°C. After incubation, oocytes were returned to the electrophysiology rig and washed for 10 min in recording buffer, and glycine concentration–responses were repeated. Currents were fit to the modified Michaelis–Menten equation (see the Materials and Methods section) and normalized to the $V_{max}$ of baseline responses. Data information: in (A, B), data points are presented as the mean ± SEM (n ≥ 5).

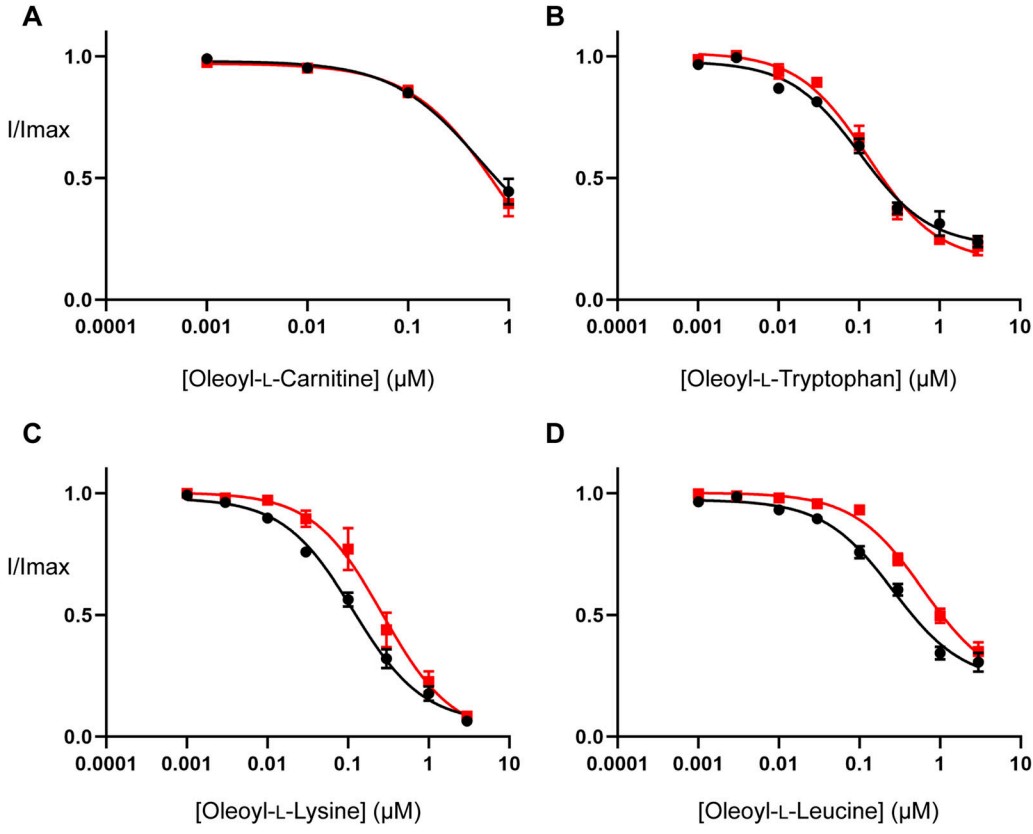

**Figure 6. Cholesterol depletion reduces the potency of some bioactive lipids at WT GlyT2 expressed in *Xenopus laevis* oocytes.**
**(A, B, C, D)** $EC_{50}$ concentration of glycine was co-applied with increasing concentrations of (A) oleoyl-L-carnitine, (B) oleoyl-L-tryptophan, (C) oleoyl-L-lysine, and (D) oleoyl-L-leucine ranging from 1 nM to 3 µM. Concentration–responses performed on control (black) and cholesterol-depleted (red) oocytes are shown. Cholesterol depletion was performed by incubating oocytes in 15 mM M$\beta$CD for 30 min at 32°C. After M$\beta$CD incubation, oocytes were moved to the recording chamber and washed in recording buffer for 10 min before commencing bioactive lipid concentration–responses. Bioactive lipids were applied cumulatively at room temperature with progression to higher concentrations occurring once a stable current was achieved after administration of the previous concentration. Data information: in (A, B, C, D), raw currents were normalized to currents generated by application of the glycine $EC_{50}$ alone. Data points are presented as the mean ± SEM (n ≥ 5).

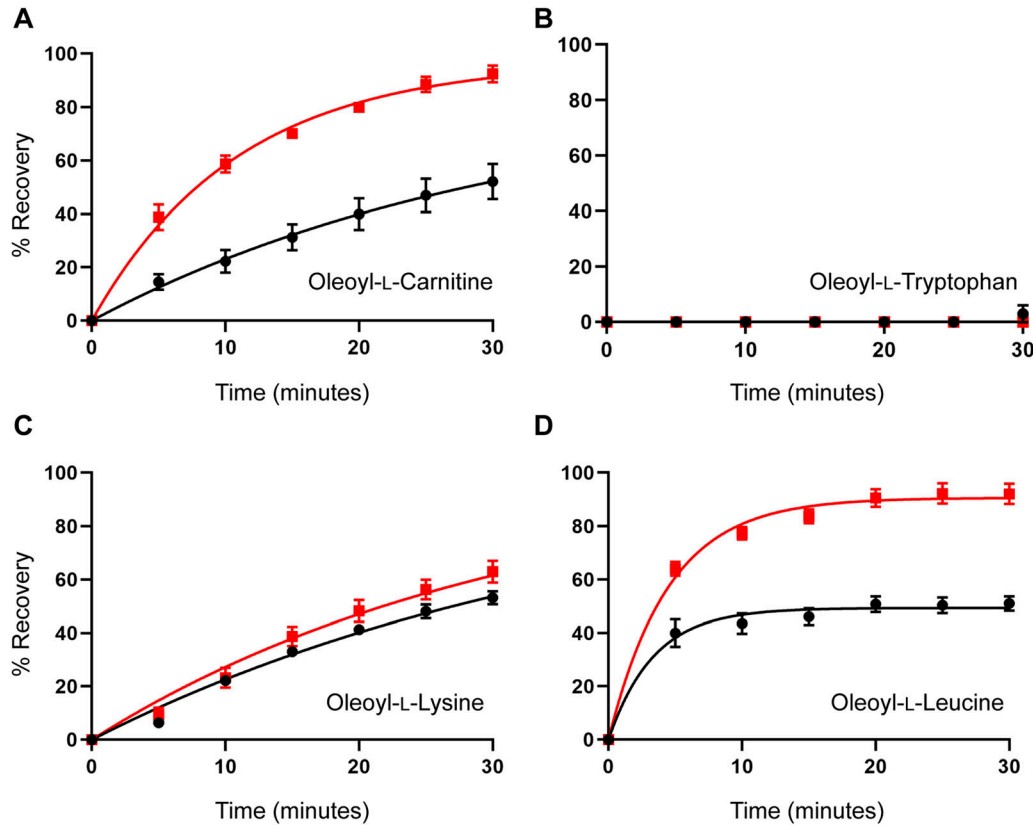

**Figure 7.  Cholesterol depletion enhances the reversibility of inhibition of WT GlyT2 expressed in *Xenopus laevis* oocytes by some bioactive lipids.**
**(A, B, C, D)** Washout time course of (A) oleoyl-L-carnitine, (B) oleoyl-L-tryptophan, (C) oleoyl-L-lysine, and (D) oleoyl-L-leucine. Membrane cholesterol was depleted by treating oocytes with 15 mM M$\beta$CD at 32°C for 30 min. After a 10-min wash period in recording buffer, an EC$_{50}$ concentration of glycine was co-applied with an IC$_{50}$ concentration of inhibitor for 4 min. After exposure to inhibitors, the EC$_{50}$ of glycine was reapplied at 5-min intervals for 30 min. % Recovery responses are shown for control (black) and M$\beta$CD-treated (red) oocytes. Data information: in (A, B, C, D), raw currents were normalized to the currents generated by application of the glycine EC$_{50}$ alone. Data points are presented as the mean ± SEM (n ≥ 5).

maximal recovery with sufficient time. Thus, a greater level of transport current recovery at the same timepoint suggests cholesterol depletion reduces the recovery half-life, allowing OLCarn to wash out quicker. In contrast, no difference was observed in the shape of the curves, or transport current recovery after 30 min, after OLLys application, indicating cholesterol depletion does not alter its reversibility (Fig 7 and Table S8). Together, these data suggest that interactions with membrane cholesterol influence the reversibility of some, but not all, bioactive lipids.

### Mutagenesis of the CHOL1 site mimics cholesterol depletion effects

In previous studies on SERT, it has been demonstrated that mutations in the cholesterol binding site destabilize the outward-facing state of the transporter and reduce the affinity for serotonin (Laursen et al, 2018). The MD simulations of GlyT2 suggest that cholesterol binds to a similar site to that of SERT and also suggest that cholesterol may influence inhibitor interactions with the transporter. Notably, in these simulations, cholesterol was recruited from the inner leaflet to CHOL1, then flipped within the CHOL1 site such that the 3′ hydroxyl and sterol ring system was bound to CHOL1 and the isooctyl tail oriented towards the

membrane. To confirm the nature of cholesterol interactions with GlyT2 and their functional consequences, site-directed mutagenesis was performed on residues that maintained the longest interactions throughout MD simulations, namely, Y430 (Y430F/L), T508 (T508I), T512 (T512A), and F515 (F515V/W) (Fig 8). To examine the importance of cholesterol orientation, L198 (L198A) and I201 (I201V/N) were also mutated. The impacts of these mutations were investigated by analysing the rates of glycine transport and the effects on inhibition of glycine transport.

With the exception of I201V/N and T508I, all GlyT2 mutants produced reliable and robust glycine-dependent transport currents (Fig 9 and Table S9). The TM5 mutants Y430F/L did not alter glycine transport kinetics; however, the kinetics were altered by the remaining mutants. These changes align with what has been reported for hSERT kinetics after mutagenesis of the corresponding residues, which have been proposed to occur through direct and indirect destabilization of non-flipped cholesterol in CHOL1 (Laursen et al, 2018). When compared to WT GlyT2, L198A increased the glycine EC$_{50}$ 1.5-fold from 19 to 28 $\mu$M ($P$ = 0.0097). Given the proximity of L198 to the 3′ hydroxyl of cholesterol in the non-flipped orientation (Fig 8), this reduced apparent affinity may reflect a destabilization of cholesterol at the CHOL1 site, favouring an inward-facing GlyT2 conformation. The remaining mutated residues

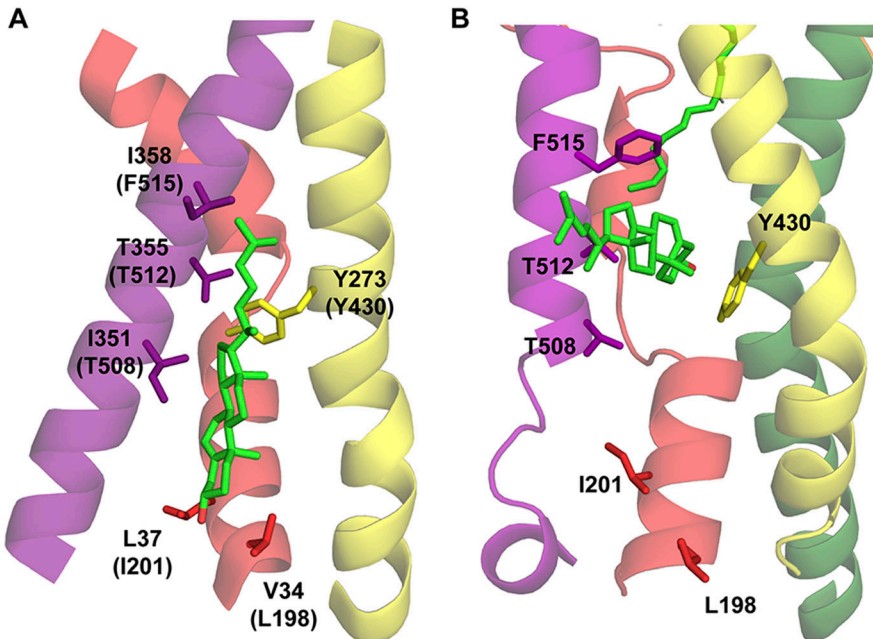

**Figure 8. Cholesterol coordinating residues investigated by site-directed mutagenesis.**
**(A, B)** Representative snapshot from MD simulations showing key stabilizing residues of cholesterol in (A) the non-flipped orientation as observed previously in the dDAT structure (PDB ID: 4M48), and (B) the flipped orientation observed in MD simulations are represented as sticks. Residues in (A) are labelled using the dDAT numbering with the corresponding GlyT2 numbering in brackets. Residues in (B) are labelled using the GlyT2 numbering. Important transmembrane domains shown are TM1 (red), TM5 (yellow), TM7 (purple), and TM8 (green). Cholesterol shown as green sticks with oxygen atoms in red. Note the proximity of the cholesterol hydroxyl group to the coordinating residues in each orientation.

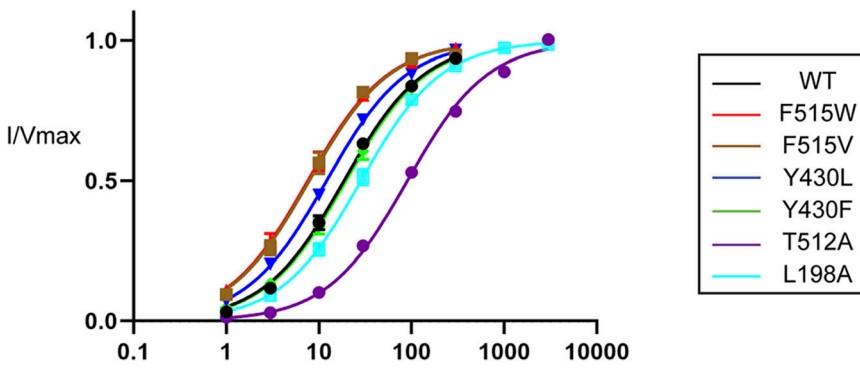

**Figure 9. Glycine-dependent currents of WT and CHOL1 mutant GlyT2 transporters expressed in *Xenopus laevis* oocytes.**
The transport activity of WT and mutant GlyT2 transporters was determined by measuring glycine-dependent currents in response to increasing concentrations of glycine from 1 μM to 3 mM. Individual glycine concentrations were continuously applied until a stable current was achieved, after which point glycine was completely removed by washing in recording buffer before proceeding to the next concentration. Currents were fit to the modified Michaelis–Menten equation (see the Materials and Methods section). Data information: raw currents were normalized to the $V_{max}$ for each cell. Data points are presented as the mean ± SEM with n ≥ 3 from at least two batches of oocytes.

are in close proximity to cholesterol, either to the isooctyl tail in the non-flipped orientation or to the rigid sterol ring system in the flipped cholesterol orientation (Fig 8). We hypothesize that these mutations could alter cholesterol binding either by a direct mechanism after it adopts a flipped orientation or by an indirect mechanism before cholesterol flipping. For example, T512A generated a 4.8-fold reduction in apparent glycine affinity exhibiting an $EC_{50}$ of 90 $\mu M$ ($P < 0.0001$). In hSERT, the equivalent residue T371 interacts with V367 and Y289 (T508 and Y430 in GlyT2) and mutation to an alanine was proposed to destabilize the positioning of these residues, impeding cholesterol binding and favouring the inward-facing transporter conformation (Laursen et al, 2018). Therefore, the GlyT2 T512A mutation presented here may perturb binding of non-flipped cholesterol through a similar mechanism, which would shift the transporter equilibrium and reduce the apparent substrate affinity. In contrast, F515V and F515W enhanced apparent glycine affinity, shifting the $EC_{50}$ to 7.8 $\mu M$ ($P = 0.0147$) and 7.6 $\mu M$ ($P = 0.0034$),

respectively. As F515 is located one helical turn above T512, it may also be altering cholesterol binding via an indirect mechanism.

If cholesterol binding helps to stabilize the lipid inhibitors in the LAS, we would expect that mutations that disrupt cholesterol binding may mimic the effects of cholesterol depletion and influence inhibitor activity. The inhibitory activity of bioactive lipids was examined by measuring concentration–responses on CHOL1 mutant transporters. All mutant transporters affected the activity of various bioactive lipids except F515V and L198A (data not shown). Consistent with observations under cholesterol depletion conditions, the inhibitory activity of OLCarn was unaffected by the majority of CHOL1 mutants except Y430F (Fig 10 and Table S10). Application of 1 $\mu M$ OLCarn produced significantly greater inhibition of Y430F compared with WT, increasing the level of inhibition by 15.5% ($P = 0.0156$). Although no differences in $IC_{50}$ values were found for these transporters, the increased level of inhibition likely reflects an increase in potency rather than efficacy as these curves do

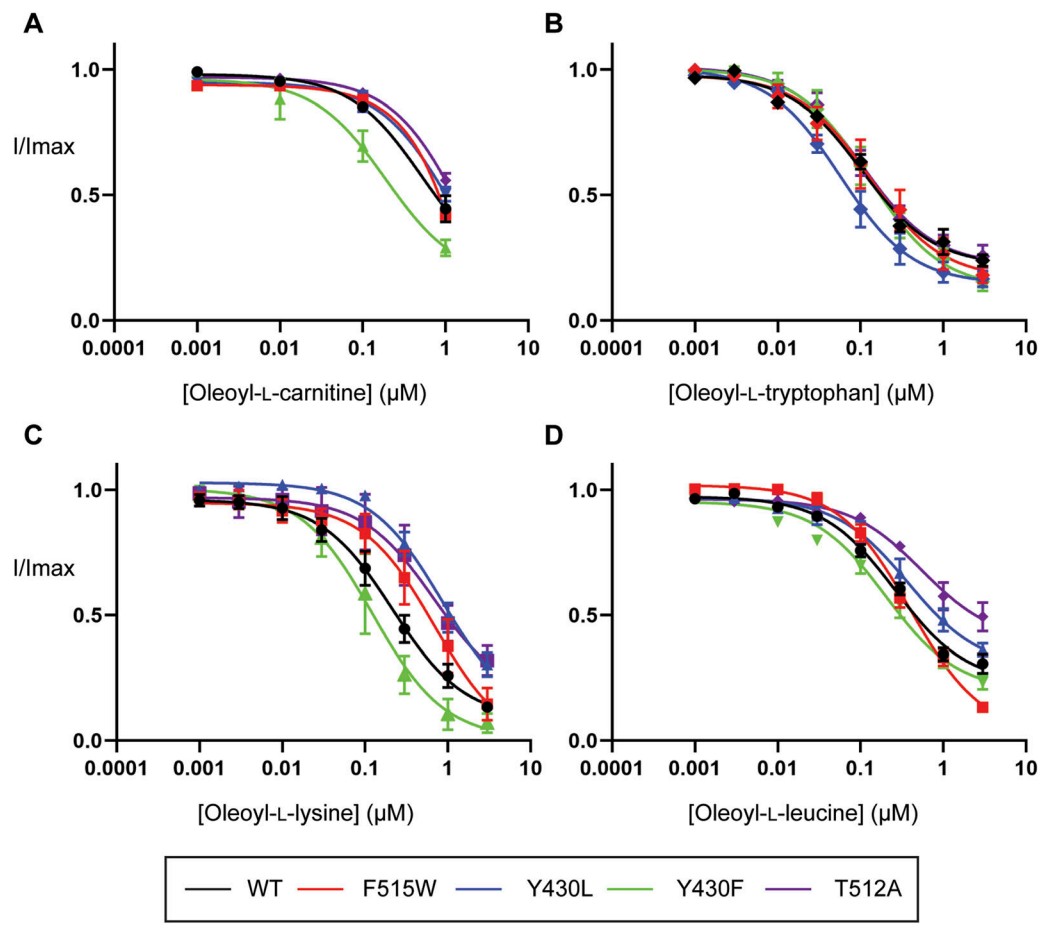

**Figure 10. Inhibitory activity of bioactive lipids at WT and CHOL1 mutant GlyT2 transporters expressed in *Xenopus laevis* oocytes.**
**(A, B, C, D)** $EC_{50}$ concentration of glycine was co-applied with increasing concentrations of (A) oleoyl-L-carnitine, (B) oleoyl-L-tryptophan, (C) oleoyl-L-lysine, and (D) oleoyl-L-leucine ranging from 1 nM to 3 µM. Bioactive lipids were applied cumulatively at room temperature with progression to higher concentrations occurring once a stable current was achieved after administration of the previous concentration. Data information: raw currents were normalized to currents generated by application of the glycine $EC_{50}$ alone. Data points are presented as the mean ± SEM with n ≥ 5 from at least two batches of oocytes.

not plateau, limiting determination of the $IC_{50}$. More refined $IC_{50}$ values could not be determined because of membrane disruptive effects observed with higher concentrations of OLCarn preventing their characterization. Changes in the inhibitory activity of other bioactive lipids also aligned well with the cholesterol depletion experiments. The activity of OLTrp, which is not predicted to interact with cholesterol and was unaffected by cholesterol depletion, was also not altered by any of the mutations (Fig 10 and Table S10). In contrast, OLLys and OLLeu, both of which had reduced potency after cholesterol depletion, were affected by CHOL1 mutants (Fig 10 and Table S10). The potency of OLLys was reduced fourfold from 220 nM on WT to 860 nM on Y430L ($P$ = 0.0407). In addition, the inhibition at 3 µM was reduced by 16.9% ($P$ = 0.0148), indicating that the actual shift in potency is likely to be larger if higher concentrations could be tested to allow the curve to plateau at its true maximum. However, because of the possibility of micelle formation limiting the free concentration of the bioactive lipids at higher concentrations we were unable to test concentrations >3 µM. The potency of both OLLys and OLLeu was shifted by the T512A mutation as inferred by reductions in the level of inhibition by 18.6% ($P$ = 0.0073)

and 35.9% ($P$ = 0.0053), respectively. In contrast to the T512A mutation, application of 3 µM OLLeu produced significantly greater inhibition of F515W, increasing by 17.3% ($P$ = 0.0098), suggesting an increase in potency.

The MD simulations suggest that cholesterol bound in the CHOL1 site stabilizes some of the lipids bound to the LAS. To characterize the effect of CHOL1 mutants on the reversibility of bioactive lipids, washout assays were performed as described above. No recovery of current was observed after application of OLTrp to any of the CHOL1 mutants (Fig 11 and Table S11), which is consistent with the observation that membrane cholesterol does not modulate the activity of this bioactive lipid. In contrast, the reversibility of OLCarn, OLLeu, and OLLys was significantly altered by the CHOL1 mutations (Fig 11). The reversibility of OLCarn was enhanced with T512A, compared to WT GlyT2, with recovery after 30 min increased 1.8-fold from 45.7% to 79.4% ($P$ < 0.0001). F515W also significantly increased the recovery of OLCarn at 30 min by 1.3-fold to 60.1% ($P$ = 0.0432), whereas it was unaltered by F515V. Intriguingly, the TM5 mutants Y430F/L had opposing effects on the recovery of OLCarn. Reducing the steric bulk in the region of Y430 via mutation to a leucine (Y430L)

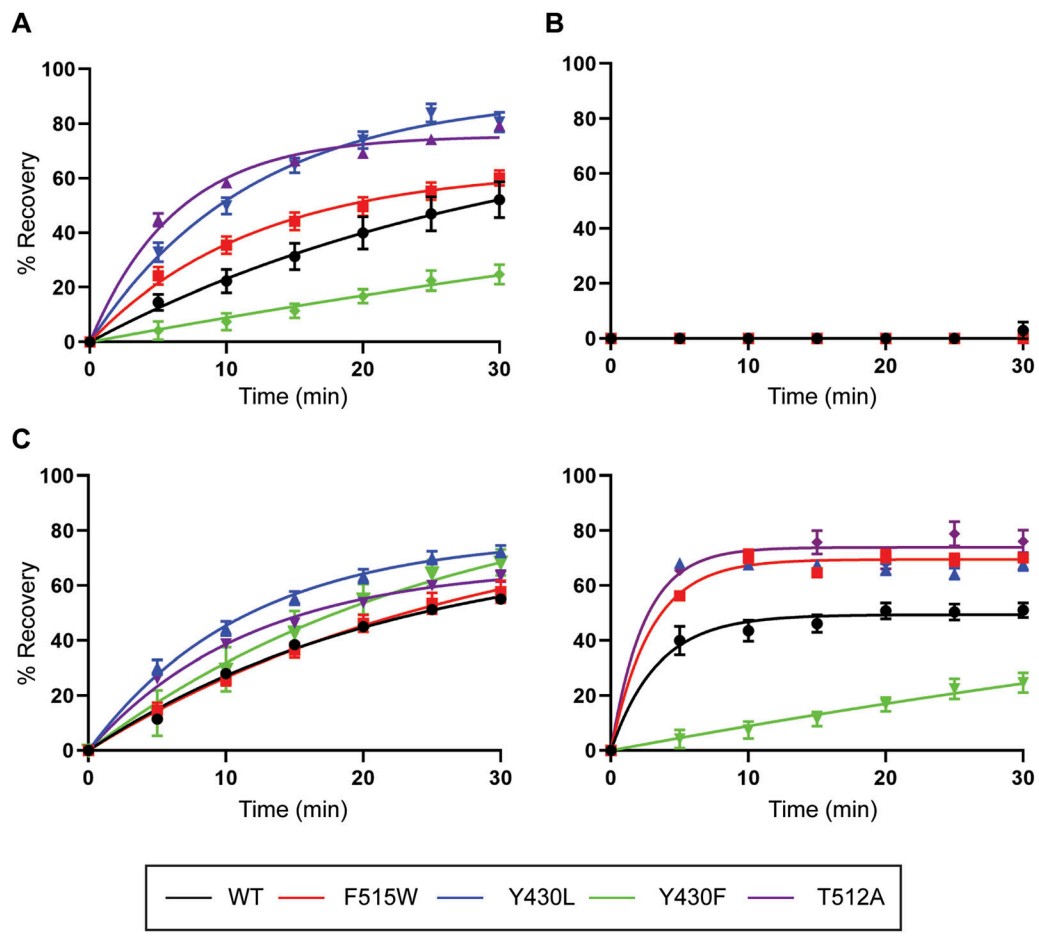

**Figure 11. Reversibility of bioactive lipid inhibition of WT and CHOL1 mutant GlyT2 transporters expressed in *Xenopus laevis* oocytes.**
**(A, B, C, D)** Washout time course of (A) oleoyl-L-carnitine, (B) oleoyl-L-tryptophan, (C) oleoyl-L-lysine, and (D) oleoyl-L-leucine. Washouts were performed by co-applying an IC$_{50}$ concentration of inhibitor with an EC$_{50}$ concentration of glycine to *Xenopus laevis* oocytes expressing WT and mutant GlyT2 transporters for 4 min. After exposure to the inhibitors, the EC$_{50}$ of glycine was reapplied at 5-min intervals for 30 min. Data information: raw currents were normalized to currents generated by application of the glycine EC$_{50}$ alone. Data points are presented as the mean ± SEM (n ≥ 5).

resulted in significantly enhanced recovery of OLCarn after 30 min to 80.6% (*P* < 0.0001). In contrast, conserving the steric bulk but removing the polarity of the hydroxyl group via mutation to a phenylalanine (Y430F) produced a significant reduction in reversibility of OLCarn with only 24.7% (*P* = 0.0002) recovery of current achieved after 30 min. This result highlights the potential significance of the aromatic ring–mediating interactions with bound cholesterol. Changes in OLLeu reversibility aligned well with what was observed for OLCarn (Fig 11 and Table S11). Specifically, F515W, Y430L, and T512A increased the recovery of current 30 min after OLLeu application by 19.0% (*P* = 0.0001), 16.8% (*P* = 0.0004), and 25.1% (*P* < 0.0001), respectively, whereas Y430F reduced recovery by 24.5% (*P* < 0.0001). In addition, as the recovery from OLLeu inhibition reached a stable plateau on each of the transporters tested, the half-life was also able to be determined. No changes were observed in the half-life recovery of OLLeu on any of the mutants (Table S11), suggesting a disrupted cholesterol interaction increases the total recovery but does not alter the rate at which that recovery is achieved. Lastly, both the Y430L and Y430F mutations increased the recovery of OLLys after 30 min by 17.4% (*P* = 0.0004) and 13.4%

(*P* = 0.0044), respectively (Fig 11 and Table S11). The enhanced reversibility of OLLys at Y430F contrasts with observations for other lipids and may result from the altered interactions between Y430F and cholesterol because of changes in cholesterol orientation in the presence of OLLys.

To understand the molecular basis for the effect of GlyT2 mutants on inhibitor activity (F515W, T512A, Y430F, and Y430L), triplicate 500-ns atomistic MD simulations were performed with OLCarn bound to the LAS and a cholesterol molecule in the CHOL1 binding site. Regardless of the mutation, little change is observed in the positioning of OLCarn in the LAS (Fig 12 and Table S12); however, there were changes in the interaction between bound cholesterol and the nearby residues forming the boundary between the LAS and CHOL1 (Table S13). In the T512A mutant, the loss of steric bulk increased the mobility of bound cholesterol but did not significantly change the distribution of residues that interact with cholesterol (i.e., V205, Y430, F515, A208, and V434; Fig 12 and Table S12). In the combined 1,500-ns simulation of F515W, cholesterol remains bound for a greater portion of the total simulation time than observed for the WT system (86% versus 66%, respectively). In one

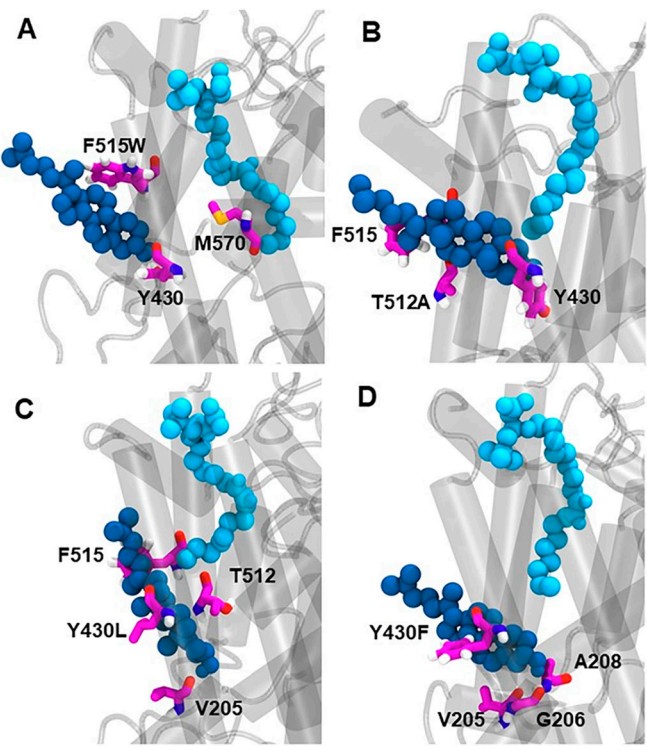

**Figure 12. Interaction between bound cholesterol, CHOL1 binding site residues, and oleoyl-L-carnitine in the four GlyT2 CHOL1 site mutants.**
**(A, B, C, D)** Representative snapshots from MD simulations in which oleoyl-L-carnitine (sky blue) occupies the LAS in simulations of the (A) F515W, (B) T512A, (C) Y430L, and (D) Y430F GlyT2 mutants. Cholesterol is in dark blue, and key CHOL1 residues are fuchsia.

replicate of the F515W system, the bound cholesterol moved out of the binding site, such that the polar head of cholesterol interacted with the heterocyclic amine of F515W (Table S13). When cholesterol exited the CHOL1 binding site, a simultaneous rotation of M570 blocked the hydrophobic interactions between cholesterol and the lipid inhibitor. Contacts between the CHOL1 residue V205 and Y430 were maintained (Table S13) (Fig 12).

In WT GlyT2 simulations, cholesterol interacted with Y430 for 70% of the total simulation time (Table S13). When Y430 is mutated to leucine (Y430L), the total contact time is reduced to 63%. The reduced contact time can be attributed to the loss of steric bulk on mutation to leucine, which reduced the coordination of cholesterol within the cavity and significantly increased its mobility. In the Y430L simulations, cholesterol reoriented to align parallel to OLCarn (Fig 12). The reoriented cholesterol maintains contact with T512, F515, and V205 observed in the WT simulations, and forms new contacts with G206, Y207, and L433 (Table S13). In contrast, the Y430F mutation increased hydrophobicity while maintaining steric bulk. Cholesterol maintained contact with Y430F for 98% of the trajectory, pointing to the significance of the aromatic interaction. Cholesterol was observed to bind deeper into the binding pocket (Fig 12), forming new contacts with G206 and A208 near the conserved V205 contact. Similarly, contacts were formed with V434 near F430, A511 and T512 near F515. This subsequently increased the hydrophobicity and stability of the OLCarn binding site, reducing the repulsion of the alkyl tail.

## Discussion

In previous studies, we have shown that bioactive lipids bind to the LAS of GlyT2 and act as inhibitors of substrate transport (Mostyn et al, 2019b; Wilson et al, 2021a). Here, we show that the allosteric inhibition of substrate transport by lipid inhibitors at the GlyT2 LAS is modulated by the recruitment of membrane cholesterol from the vicinity of the CHOL1 site. The recruitment mechanism involves the flipping of cholesterol from its orientation in the membrane, and the subsequent insertion of the cholesterol 3′ hydroxyl group into a cholesterol binding cavity formed between TM1, TM5, and TM7, followed by its interaction with the base of the LAS and bound inhibitor. The sensitivity and potency of inhibition correspond to the interaction of the cholesterol 3′ hydroxyl group with the acyl chain terminus of the lipid inhibitor. Of the four different lipid inhibitors investigated, inhibitors that protruded deeper into the LAS altered the relative orientation of the Y430 (TM5) side chain to create a cavity that membrane cholesterol occupying the CHOL1 site enters and binds to. We show that depletion of membrane cholesterol, and mutation of the residues in the CHOL1 site, also reduces the efficacy of lipid inhibitors bound to the LAS, supporting the cholesterol modulation of inhibitor activity. Previous studies have shown that application of the cholesterol-sequestering agent, MβCD, alters the activity and reversibility of the GlyT2 lipid inhibitor OLCarn (Carland et al, 2013). Although this effect was originally attributed to a direct interaction between MβCD and OLCarn, the potential role of cholesterol was not explored. We propose that the binding of cholesterol to the CHOL1 site stabilizes the outward-facing conformation of GlyT2 to facilitate the formation of an accessible LAS and the subsequent binding of lipid inhibitors. Our CG MD simulations indicate that the lipid inhibitors access the LAS directly from the extracellular solution, rather than from the membrane. In addition, once in the LAS, the mechanism of inhibition appears to be modulated by cholesterol from the CHOL1 site binding between TM1, TM5, and TM7. We hypothesize that the deep penetration of lipid inhibitors into the LAS is critical for the interaction of cholesterol with the lipid inhibitor, which together may further stabilize GlyT2 in an outward-open conformational state, inhibiting the transport process. Therefore, the more pronounced effect of cholesterol depletion on the cholesterol-dependent inhibitors OLLys, OLLeu, and OLCarn, compared with the cholesterol-independent inhibitor OLTrp, may be due to the deeper penetration of the lipid tail, which facilitates a direct interaction between the lipid inhibitor and cholesterol.

Cholesterol has long been implicated in the regulation of the NSSs of the SLC6 transporter family, however, until recently the mechanism has been largely unknown (Scanlon et al, 2001). Recent structural data, simulations, and experimental studies have identified five conserved cholesterol binding sites in the outward-open conformations of homologous NSS transporters (Laursen et al, 2018; Zeppelin et al, 2018; Wilson et al, 2021b). Of these cholesterol binding sites, binding of cholesterol to the CHOL1 site has been shown to shift the transporter conformational equilibrium, stabilizing the outward-open conformation of DAT and SERT (Laursen et al, 2018; Zeppelin et al, 2018). Furthermore, membrane cholesterol depletion has been shown to decrease the activity of

modulators that bind to the outward-open conformation and increase the activity of modulators that bind to the inward-open conformation (Scanlon et al, 2001; Laursen et al, 2018), supporting the proposed role of cholesterol in stabilizing the outward-open conformation.

Cholesterol is heterogeneously distributed in neuronal cells and constitutes 44.6 mol % of the neuronal membrane (Ingólfsson et al, 2017; Wilson et al, 2021b). Plasma cholesterol is transported in lipoproteins, which cannot cross the blood–brain barrier. As a result, cholesterol homeostasis is independent of plasma cholesterol levels and neuronal cholesterol is synthesized in situ (Zhang & Liu, 2015) and its concentrations are closely regulated in normal brain function (Zhang & Liu, 2015). Defects in brain cholesterol metabolism and homeostasis have been shown to be implicated in psychiatric and neurodegenerative diseases (Zhang & Liu, 2015). Because neuronal cholesterol is synthesized in situ, cholesterol-lowering agents such as statins impact neuronal cholesterol biosynthesis (Fracassi et al, 2019). Further studies have shown that statins can be detected at significant levels in the brain after a single dose (Johnson-Anuna et al, 2005; Fracassi et al, 2019) and that long-term treatment with the statin simvastatin reduces the cholesterol content of membrane lipid rafts and significantly reduces cholesterol levels in the extracellular leaflet of brain cells, altering membrane properties such as fluidity (Kirsch et al, 2003). These observations, together with the results in this study, suggest that the actions of brain-penetrating statins may influence glycine transport and reduce the efficacy of glycine transport inhibitors. Thus, the future development of GlyT2 inhibitors for the treatment of neuropathic pain may need to consider their effectiveness in patients being treated with statins.

Modulation of neurotransmitter transporter function underpins the management of a number of neurological conditions, including depression, anxiety, epilepsy, Parkinson's disease and addiction, and chronic pain (Pramod et al, 2013). When considered together, the previous data for SERT and DAT (Laursen et al, 2018; Zeppelin et al, 2018), and the results for GlyT2 presented here, strongly suggest that variations in membrane cholesterol levels influence the conformational equilibria of neurotransmitter transporters during the transport cycle, and therefore the physiological clearance of neurotransmitters from the synaptic cleft. Furthermore, our results indicate that membrane cholesterol may impact the activity and efficacy of inhibitors of GlyT2 and other homologous neurotransmitter transporters. Specifically, depletion of cholesterol from membrane lipid rafts may reduce the binding of cholesterol to the CHOL1 site, shifting the conformational equilibrium to the inward-open conformation reducing the accessibility of both the substrate binding site and the LAS. A failure to respond to SSRIs, SNRIs, and competitive inhibitors of neurotransmitter transporters is notable in some patients; one of the factors that may contribute to the lack of efficacy could be related to membrane cholesterol regulation.

The interaction between cholesterol in the CHOL1 site and bound inhibitors in the LAS provides a promising and novel avenue for optimization of pharmaceutical designs that will promote both a selective and reversible inhibition of GlyT2 to treat chronic pain. The observations that cholesterol can influence the equilibrium between inward- and outward-facing states suggest that the process could also be pharmacologically manipulated by derivatives of cholesterol—such as the various neurosteroids (allopregnanolone), which would be analogous to the neurosteroid manipulation of ligand-gated ion channels. Drugs may be developed that mimic the effects of cholesterol. Furthermore, chronic pain is associated with increased levels of reactive oxygen species and oxidative stress (Hassler et al, 2014; Hendrix et al, 2020). Cholesterol is more susceptible to oxidation by reactive oxygen species than phospholipids because of the 6-double bond and the vinylic methylene group at C-7 in the B ring of the styrene, and the isooctyl side chain at C-17 (Murphy & Johnson, 2008), and oxysterols are important regulators of neuronal lipid homeostasis (Zhang & Liu, 2015). As neuronal membrane oxysterol levels can increase to up to 20 mol % in pathological conditions, competitive binding between oxysterols and cholesterol at the GlyT2 CHOL1 site must also be considered in a personalized medicine approach for pharmacological design of GlyT2 inhibitors of the ascending pain pathway.

Another implication of the results presented in this study is whether a similar allosteric ligand binding site exists in other closely related SLC6 transporters for dopamine, serotonin, and GABA. If this site is present on these other transporters, it may provide the opportunity to develop new classes of neurotransmitter transport inhibitors for the treatment of a range of neurological disorders such as depression, drug addiction, Parkinson's disease, and schizophrenia.

# Materials and Methods

## Molecular dynamics

### CG simulations

The GROMACS 2016.1 MD engine (Abraham et al, 2015) and the MARTINI 2.2P forcefield (de Jong et al, 2013) were used for all CG simulations. The previously published homology model of GlyT2 with two $Na^+$ ions and the substrate glycine bound was used as a starting structure for modelling (Subramanian et al, 2016; Carland et al, 2017). GlyT2 was converted to a GoMARTINI CG model using the *go_martinize* package with Gō contacts of 9.414 kJ/mol (Poma et al, 2017). The substrate ions were manually converted to CG MARTINI 2.0 ions. The protein and its substrates were embedded in an 80% POPC and 20% cholesterol (CHOL) bilayer and solvated with polarizable (MARTINI 2.2P) CG water (box size: 166 × 194 × 160 Å) using insane.py (Wassenaar et al, 2015). The bilayer was modelled with MARTINI 2.0 lipids. The system was solvated with the MARTINI polarizable water model (Yesylevskyy et al, 2010). To replicate the patch-clamping experiments where the additives are only applied to the extracellular side of the cell and simulate spontaneous binding of the lipid inhibitors to the extracellular leaflet, a layer of position-restrained water was created by selecting a single layer of water molecules ~10 Å below the intracellular leaflet. The layer of position-restrained water was extended in the x- and y-directions to ~1.5 Å from the box edges to allow for changes in the box size upon simulation. The water in this layer was restrained with a force constant of 1,000 kJ $mol^{-1}$ $nm^{-2}$ for the duration of the simulations. 20 lipid inhibitors of a single type, namely, OLLys, OLLeu, OLSer, or OLTrp, were placed at random coordinates within the extracellular

solution or between the layers of the position-restrained water and box edge. The net charge on each system was neutralized, and NaCl was added to a concentration of 150 mM to mimic physiological conditions.

The entire system was minimized and then equilibrated with a force constraint on the protein backbone. The constraint on the protein backbone was slowly reduced through five sequential simulations of 1 ns each, with constraint weights of 1,000, 500, 100, 50, and 10 kJ mol$^{-1}$ nm$^{-2}$, respectively. Each system was then simulated for 10 $\mu$s in triplicate without constraints on the protein. A new random starting velocity was assigned at the start of each replicate simulation. Coordinates were saved every 20 ns, and a time step of 5 and 20 fs was used for the equilibration and production simulations, respectively. All simulations were performed in the NPT ensemble. The simulations were performed at 310 K and 1 bar with the velocity-rescale thermostat ($\tau_T$ = 1 K) and Berendsen barostat ($\tau_P$ = 3.0 bar). The pressure was modelled with semi-isotropic pressure coupling (isothermal compressibility = 3.0 × 10$^{-4}$ bar$^{-1}$), with the pressure being isotropic in the plane of the bilayer. The length of covalent bonds was constrained using the LINCS algorithm. Electrostatic interactions were calculated using the reaction-field method, and van der Waals interactions were calculated with a cut-off of 11 Å.

### Backmapped atomistic simulations

Atomistic simulations were performed using the GROMACS 2016.1 MD package (Abraham et al, 2015) and the GROMOS 54a7 forcefield (Schmid et al, 2011). Lipid inhibitors were modelled using parameters generated using the Automated Topology Builder and Repository (Stroet et al, 2018) (OLLys MoleculeID: 252919, OLTrp MoleculeID: 252930, OLLeu MoleculeID: 252921, and OLCarn MoleculeID: 296337), as previously described (Wilson et al, 2021a). To ensure that there was no isomerization around the cis double bond, the force constant related to this dihedral angle was adjusted from 5.86 to 41.80 kJ/mol/rad$^2$, as previously described (Wilson et al, 2021a). The water model was SPC (Berendsen et al, 1987). To further understand the binding of the lipid inhibitors to GlyT2, selected frames from simulations in which OLLeu was bound at the LAS and CHOL was interacting with the CHOL1 site were backmapped to atomistic coordinates using backmap.py (Wassenaar et al, 2014). Only lipid inhibitors within 10 Å of the protein were retained in atomistic simulations. All water was treated as unrestrained. This local region was then embedded within an 80% POPC/20% CHOL bilayer and solvated such that the final system size was 166 × 194 × 100 Å. The system was minimized and then equilibrated with a harmonic constraint on the protein CA atoms. The harmonic constraints on the CA atoms were slowly reduced over five sequential 1-ns simulations from 1,000, to 500, 100, 50, and finally 10 kJ mol$^{-1}$ nm$^{-2}$. Each system was then simulated for 500 ns in triplicate without constraints with a 2-fs time step. A new random starting velocity was assigned at the start of each replicate simulation. Coordinates were saved every 0.1 ns. All simulations were performed in the NPT ensemble. The simulations were performed at 310 K and 1 bar with the velocity-rescale thermostat ($\tau_T$ = 0.1 K) and Berendsen barostat ($\tau_P$ = 0.5 bar). The pressure was modelled with semi-isotropic pressure coupling (isothermal compressibility = 4.5 × 10$^{-5}$ bar$^{-1}$), with the pressure being isotropic in the plane of the

bilayer. The length of the covalent bonds was constrained using the LINCS algorithm. Electrostatic interactions were calculated using the particle mesh Ewald, and van der Waals interactions were calculated with a cut-off of 10 Å.

### Atomistic simulations of GlyT2 with docked inhibitor

The equilibrated backmapped system was used as a starting point for the atomistic simulations with OLLys, OLCarn, and OLTrp bound to the LAS. OLLeu, solvent, and ions were removed from the backmapped simulation. The conformation of the lipid inhibitor in the LAS was obtained by overlaying the final conformation obtained from previous simulations performed with the lipid inhibitor docked into the LAS (Mostyn et al, 2019b). Each system was then solvated, energy-minimized, equilibrated, and simulated using the protocol outlined above for the backmapped atomistic simulations.

### Atomistic simulations of GlyT2 mutations

The equilibrated backmapped system was again used as a starting point for atomistic simulations incorporating the effect of GlyT2 mutations on inhibitor and CHOL binding. PyMOL (version 2.5.0; The PyMOL Molecular Graphics System, Schrödinger, LLC) was used to incorporate a series of individual GlyT2 point mutations (T512A, Y430L, Y430F, and F515W). To characterize the effect of each mutation on inhibitor and CHOL binding, only one mutation was included per simulation system. Each system was then solvated, energy-minimized, equilibrated, and simulated using the protocol outlined above for the backmapped atomistic simulations.

### Atomistic simulations of inward-facing GlyT2

A homology model of the inward-facing conformation of GlyT2 was generated using SWISS-MODEL (Waterhouse et al, 2018). Here, the inward-facing cryoEM structure of human SERT (PDB ID: 6DZZ) (Coleman et al, 2019) was used as a template. The protein was embedded in an 80% POPC and 20% cholesterol bilayer. The inward-facing homology model was then solvated, energy-minimized, equilibrated, and simulated using the protocol outlined above for the backmapped atomistic simulations.

### Simulation analysis

Area per lipid and bilayer thickness were calculated using FATSLiM (Buchoux, 2017). Simulations were visualized and contact residues calculated using VMD 1.9.3 (Humphrey et al, 1996). The pocket volume for the LAS was calculated using trj_cavity (Paramo et al, 2014).

## Cholesterol depletion and electrophysiology

### Generation of WT and mutant RNA encoding GlyTs

GlyT2a (referred to herein as GlyT2) was subcloned into the plasmid oocyte transcription vector (pOTV) for expression in *Xenopus laevis* oocytes. Oligonucleotide primers incorporating desired mutations were synthesized by Sigma-Aldrich and used to generate point mutations in the GlyT2 cDNA/pOTV construct using the Q5 site-directed mutagenesis kit (New England Biolabs [Genesearch]). Amplified PCR products were transformed into chemically competent *E. coli* cells and purified using the GeneJET Plasmid Miniprep Kit (Thermo Fisher Scientific). Purified plasmid DNA was sequence-

verified by the Australian Genome Research Facility. pOTV constructs were linearized with the restriction enzyme *SpeI* (New England Biolabs [Genesearch]), and RNA was transcribed by T7 RNA polymerase using the mMESSAGE mMACHINE T7 kit (Thermo Fisher Scientific).

### Oocyte harvesting and preparation

*X. laevis* were anaesthetized with 0.17% (wt/vol) 3-aminobenzoic ethyl ester (tricaine). A single incision was made in the abdomen and a lobe of oocytes removed. Individual oocytes were separated from the follicle via digestion with 3 mg/ml collagenase A (Boehringer) at 18°C for 45–90 min. All surgeries were conducted in accordance with the *Australian Code of Practice for the Care and Use of Animals for Scientific Purposes*.

Separated stage IV/V oocytes were injected with 23–46 nl of WT or mutant RNA (Drummond Nanoject, Drummond Scientific Co.). Injected oocytes were stored in frog Ringer's solution (ND96; 96 mM NaCl, 2 mM KCl, 1 mM MgCl$_2$, 1.8 mM CaCl$_2$, and 5 mM Hepes, pH 7.5) supplemented with 2.5 mM sodium pyruvate, 0.5 mM theophylline, 50 $\mu$g/ml gentamicin, and 100 $\mu$M/ml tetracycline. Oocytes were stored at 18°C for 2–4 d until sufficient transporter levels were reached to measure transport activity. Sufficient transporter expression was defined as the onset of robust and readily reproducible inward currents in response to substrate.

### Two-electrode voltage-clamp electrophysiology

The transport activity of GlyT2 is an electrogenic process as it couples the transport of glycine to the co-transport of 3 Na$^+$ and 1 Cl$^-$ and thus can be measured using two-electrode voltage-clamp electrophysiology. Oocytes expressing GlyTs were voltage-clamped at –60 mV, and glycine-induced whole-cell currents were measured using a GeneClamp 500 amplifier (Axon Instruments), and a PowerLab 2/26 chart recorder (ADInstruments) and LabChart Software (ADInstruments).

### Membrane cholesterol depletion

Oocyte membranes were depleted of cholesterol by treating them with the cholesterol-sequestering agents M$\beta$CD and $\gamma$CD. Individual oocytes were incubated at 32°C for 30 min in 15 mM M$\beta$CD or $\gamma$CD, suspended in frog Ringer's solution. Oocytes were then washed in ND96 for 5 min, unclamped, and for a further 5 min voltage-clamped to ensure complete removal of cyclodextrin.

### Glycine concentration–responses

Functionality of WT and mutant GlyT2 transporters was assessed by applying increasing concentrations of glycine (1–3,000 $\mu$M) in ND96 until a stable current was achieved. Glycine concentrations were applied individually and completely removed from the system via washing with recording buffer before proceeding to the next concentration. EC$_{50}$ values for each transporter were determined by fitting concentration–response data to the modified Michaelis–Menten equation:

$$I = ([Gly].I_{max})/(EC_{50} + [Gly]),$$

where I is current in nA, [Gly] is the concentration of glycine, I$_{max}$ is the current generated by a maximal concentration of glycine, and

EC$_{50}$ is the concentration of glycine that generates a half-maximal current. Differences in EC$_{50}$ and V$_{max}$ values between WT and mutant transporters were analysed via a one-way ANOVA with Tukey's post hoc test. Statistical significance is presented as *$P \le$ 0.05, **$P \le$ 0.01, ***$P \le$ 0.001, and ****$P \le$ 0.0001.

### Glycine concentration–responses—cholesterol depletion

As glycine is a fully reversible substrate, glycine concentration–responses were performed on the same cell pre- and post-treatment with M$\beta$CD or $\gamma$CD. Baseline glycine concentration–responses were performed as described above and obtained before treatment with cyclodextrins (see "Membrane cholesterol depletion" in the Materials and Methods section). After cyclodextrin treatment, oocytes were returned to the electrophysiology set-up and washed for 10 min to remove cyclodextrins, before repeating glycine concentration–responses on the same cell. Glycine-dependent currents after cyclodextrin treatment were normalized to the V$_{max}$ of the control response and then fitted to the modified Michaelis–Menten equation. Differences in EC$_{50}$ and V$_{max}$ values between non-treated and cyclodextrin-treated cells were analysed via a paired two-tailed *t* test. Statistical significance is presented as *$P \le$ 0.05, **$P \le$ 0.01, ***$P \le$ 0.001, and ****$P \le$ 0.0001.

### Inhibitor concentration–response—mutagenesis

Because of the slowly reversible nature of the compounds tested, inhibitor concentration–responses were performed cumulatively. An EC$_{50}$ concentration of glycine was applied until a stable transport current was reached. EC$_{50}$ concentrations of glycine were used to produce reliable currents for each transporter, and as the inhibitors are non-competitive, application of varying glycine concentrations is unlikely to affect their activity. Increasing concentrations of inhibitors were co-applied with glycine with each concentration producing a distinct plateau before the subsequent concentration was applied. Inhibitor concentration–responses were limited to a maximum concentration of 3 $\mu$M to mitigate the risk of changes to the free drug concentration because of micelle formation. Currents after the plateau of each inhibitor concentration were normalized to the current produced by application of glycine alone and fit using the method of least squares:

$$Y = Bottom + (Top - Bottom)/(1 + 10^{[X-LogIC50]}),$$

where X is log [inhibitor], Y is current normalized to glycine in the absence of inhibitor, and Top and Bottom are the maximal and minimal plateaus, respectively. The method of least squares was constrained to have the bottom value >0 and the hill slope as –1, enabling the calculation of IC$_{50}$ values and the extent of inhibition produced by the maximal concentration applied.

IC$_{50}$ values are presented as the mean and 95% confidence interval, whereas inhibition values are presented as the mean ± SEM. Data sets are n ≥ 5 from two batches of oocytes. Statistical analysis of inhibitor concentration–responses across WT and mutant transporters was performed using a one-way ANOVA with Tukey's post hoc test for both IC$_{50}$ values and maximal inhibition. Statistical significance is presented as *$P \le$ 0.05, **$P \le$ 0.01, ***$P \le$ 0.001, and ****$P \le$ 0.0001.

### Inhibitor concentration–responses—cholesterol depletion

As the inhibitors assayed here are slowly reversible, inhibitor concentration–responses were unable to be performed on the same cell after treatment with $M\beta CD$. Thus, GlyT2 expression was tested by applying a maximal glycine concentration (300 $\mu M$) before treatment with $M\beta CD$ to deplete membrane cholesterol. After $M\beta CD$ treatment, oocytes were returned to the electrophysiology set-up and washed for 10 min in recording buffer to remove cyclodextrins. After washing, 300 $\mu M$ glycine was reapplied to the same cell and reduction in $V_{max}$ was used as an indicator of successful depletion. After successful depletion, inhibitor concentration–responses were performed as described above (see "Inhibitor concentration–response—mutagenesis" in the Materials and Methods section). For cholesterol depletion inhibitor concentration–responses, an unpaired two-tailed $t$ test was used to determine differences in $IC_{50}$ values and the inhibition produced by the maximal concentrations tested, with statistical significance represented as $*P \leq 0.05$, $**P \leq 0.01$, $***P \leq 0.001$, and $****P \leq 0.0001$.

### Inhibitor washout time course

Reversibility of GlyT2 inhibitors was determined by applying an $EC_{50}$ concentration of glycine followed by co-application of the inhibitor at an $IC_{50}$ concentration once a stable current had been achieved. Inhibitors were applied for 4 min as this was sufficient time to produce substantial inhibition for each compound and allowed for their direct comparison. At the conclusion of 4 min (time 0), ND96 buffer was perfused through the recording chamber to wash the oocyte. At 5-min intervals, up to 30 min, the $EC_{50}$ glycine concentration was reapplied, and recovery of transport was used as an indicator of reversibility. Transport currents were normalized to glycine currents in the absence of inhibitor and converted to % recovery values via

$$Y = \frac{I_t - I_{t=0}}{I_{max} - I_{t=0}},$$

where Y is % recovery, I is current in nA, $I_{max}$ is the maximal current produced by the glycine $EC_{50}$, and t is time in minutes. Washout kinetics were then determined by fitting % recovery values to the first-order rate equation:

$$Y = (Y_0 - Plateau).e^{(-Kt)} + Plateau,$$

where Y is the % recovery, $Y_0$ is the point, which crosses the Y-axis (constrained to 0) and increases to the plateau (constrained to between 0 and 100), K is the rate constant (1/t), and t is time in minutes. Differences in half-life and maximal recovery between WT and mutant transporters were analysed via a one-way ANOVA with Tukey's post hoc test. Because of the slowly irreversible nature of the inhibitors, washout time courses involving cyclodextrin treatment could not be performed on the same cell. Therefore, validation of successful cholesterol depletion was performed as above (see "Inhibitor concentration–responses—cholesterol depletion" in the Materials and Methods section) and inhibitor washout time course experiments were carried out as described earlier in this section. As these experiments could not occur on the same cell, differences in half-life and maximal recovery were analysed via an unpaired two-tailed $t$ test. In both instances,

statistical significance is presented as $*P \leq 0.05$, $**P \leq 0.01$, $***P \leq 0.001$, and $****P \leq 0.0001$.

### Quantification of cholesterol content

Control and $M\beta CD$-treated oocytes were lysed in 1% Triton X-100 (Sigma-Aldrich) by vortexing and manually disrupting the membranes with forceps. Cell lysates were centrifuged for 5 min at 17$g$ to precipitate the cell debris. The supernatant was then treated in accordance with the manufacturer's protocol for the Amplex Red Cholesterol Assay Kit (Thermo Fisher Scientific) to quantify differences in cholesterol content.

## Data Availability

Simulation parameter files, and the initial and final coordinates of the simulations are available at https://github.com/OMaraLab/GlyT2_mutants

## Supplementary Information

## Acknowledgements

This work was supported by an Australian Government National Health and Medical Research Council Project Grant APP1144429 to RJ Vandenberg and ML O'Mara. Simulations were performed with the assistance of resources and services from the National Computational Infrastructure (NCI), which is supported by the Australian Government.

### Author Contributions

ZJ Frangos: conceptualization, data curation, formal analysis, investigation, visualization, methodology, and writing—original draft, review, and editing.
KA Wilson: conceptualization, data curation, software, formal analysis, validation, investigation, visualization, methodology, and writing—original draft, review, and editing.
HM Aitken: data curation, software, formal analysis, validation, investigation, visualization, and writing—review and editing.
R Cantwell Chater: formal analysis, validation, investigation, visualization, and writing—review and editing.
RJ Vandenberg: conceptualization, resources, supervision, funding acquisition, validation, methodology, project administration, and writing—original draft, review, and editing.
ML O'Mara: conceptualization, resources, supervision, funding acquisition, validation, methodology, project administration, and writing—original draft, review, and editing.

### Conflict of Interest Statement

The authors declare that they have no conflict of interest.

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
