## [Reviewer comments · Life Science Alliance]

Life Science Alliance

Membrane cholesterol regulates inhibition and substrate transport by the glycine transporter, GlyT2

Zachary Frangos, Katie Wilson, Heather Aitken, Ryan Cantwell Chater, Robert Vandenberg, and Megan O'Mara
DOI: <https://doi.org/10.26508/lsa.202201708>

Corresponding author(s): Megan O'Mara, University of Queensland

Review Timeline:

Submission Date:	2022-09-04
Editorial Decision:	2022-10-24
Revision Received:	2022-11-07
Editorial Decision:	2022-11-25
Revision Received:	2023-01-10
Accepted:	2023-01-11

Scientific Editor: Novella Guidi

Transaction Report:

October 24, 2022

Re: Life Science Alliance manuscript #LSA-2022-01708-T

Prof. Megan L O'Mara
Australian National University
Chemistry and Molecular Biosciences
Australian National University
Brisbane, ACT 4072

Dear Dr. O'Mara,

Thank you for submitting your manuscript entitled "Membrane cholesterol regulates inhibition and substrate transport by the glycine transporter, GlyT2" to Life Science Alliance. The manuscript was assessed by expert reviewers, whose comments are appended to this letter. We invite you to submit a revised manuscript addressing the Reviewer comments.

Thank you for this interesting contribution to Life Science Alliance. We are looking forward to receiving your revised manuscript.

Sincerely,

B. MANUSCRIPT ORGANIZATION AND FORMATTING:

Reviewer #1 (Comments to the Authors (Required)):

In this paper, the authors have studied the role of cholesterol in glycine transport by the SLC6 glycine transporter GlyT2 using molecular dynamics simulations and mutagenesis of the transporter with or without cholesterol depletion. They report that the inhibition of glycine transport by bioactive lipids, which bind to the lipid allosteric site, occurs via cholesterol binding to a site consisting of transmembrane helices 1, 5 and 7. They also report that recruitment of cholesterol involves its flipping and insertion into a binding cavity where it interacts with the lipid allosteric site and the bound inhibitor. They conclude that this study may lead to the development of specific pain analgesics and may provide alternative treatment that targeting GlyT2 and other SLC6 transporters.

I find the paper potentially very interesting and the work mostly convincing. I only have a few minor comments concerning the manipulation of membrane cholesterol with cyclodextrins.

1) Fig 4 and 5 compare the transport activity of GlyT2 expressed in *X. laevis* oocytes without CD or in the presence of 15 mM CD after incubation for 30min at 32°C. Are such high CD concentrations really necessary? Aren't authors afraid that the specificity of CD towards cholesterol may be reduced at these high concentrations and high temperatures, with CD acting more like a detergent? Do authors know how much membrane cholesterol is removed by the treatment? If lower cholesterol concentrations cannot be used, it would be nice to show that effects of cholesterol depletion on OLLys and OLLeu are reverted after cholesterol re-complementation (e.g., using CD pre-loaded with cholesterol).

2) It is not clear from Legends and Methods if oocytes are pre-incubated in the presence of CD for 30min, and then further incubated in the presence of glycine (Fig 4) or bio-active lipids (Fig 5), or not. In the latter case what are incubation conditions (temperature, time)?

3) Fig 9 does not seem to be mentioned in the text.

Reviewer #2 (Comments to the Authors (Required)):

The manuscript by Frangos et al first used molecular dynamic simulations to characterize the interaction of membrane cholesterol with inhibitors of SLC6 glycine transporter GlyT2. They then used *Xenopus Laevis* Oocytes, treated or without 15 mM MCD, to experimental test the roles of cholesterol in the function of inhibitors. They also studied point-mutations in GlyT2 that potentially involved in cholesterol-inhibitor interactions on their transport activities.

Overall, the manuscript is poorly written, including abstract. This reviewer is not able to evaluate the first part of the manuscript, the molecular dynamic simulation. The review therefore will focus on experimental data and interpretations. In general, there is no direct or sufficient evidence to support that cholesterol interferes with inhibitors. Some particular problems with their experiments and interpretations are listed below.

1. For acute cholesterol depletion by MCD, more control experiments are necessary. MCD is a potent reagent to sequester cholesterol, rapidly and reliably. However, this sequestration is not entirely cholesterol specific. MCD also sequester phospholipids among other hydrophobic molecules. It is only at low concentrations (<10 mM) that MCD preferentially sequesters cholesterol. Authors here used 15 mM MCD. This makes it possible that cholesterol and phospholipids are both removed. To attribute the effect of MCD to cholesterol, it is necessary to rescue the phenotypes with cholesterol, such as cholesterol/MCD complex. Without this control experiment, the effect of MCD could not be regarded as depletion of cholesterol.

2. The effect of MCD on inhibitors are relatively minor, which would be difficult to tease out. Data is not at all well explained or interpreted. Why no MCD effect after 3uM inhibitors?

3. It is not clear without experiments mean. Kd of inhibitors with or without MCD? How would this relate to cholesterol-inhibitor interaction? The data from 4 inhibitors seems not to be coherent from Fig 4 to Fig. 6.

4. It is also unclear the purpose of the mutagenesis study. If they are aimed to illustrate the location where inhibitors interact with cholesterol, should cholesterol depletions be tested there with inhibitors?

Therefore, in second part of the manuscript, it needs much more logic and precise definition of the experimentation and interpretation, starting with controls for MCD treatment.

Both Reviewer #1 and Reviewer #2 raised comments about the conditions used in our M β CD experiments. Thus, in this section we have combined these comments and addressed them as one. Responses to Reviewer specific comments can be found later in the document.

Comment:

Are authors afraid cyclodextrin specificity may be reduced at high concentrations and temperatures.

Reply:

A variety of M β CD concentrations and incubation periods have been used to deplete cholesterol in cell systems (including oocytes). Our conditions are within the range of these published protocols (see tables below) and we have added a justification based on the need to rapidly/selectively deplete cholesterol from oocytes whilst maintaining their viability (Lines 227-231 in revised manuscript). As there can be variability in expression levels of transporters between different oocytes, we removed this variability by measuring the functional properties of GlyT2 in each oocyte both before and after M β CD.

To address the concern regarding specificity, we repeated the glycine concentration-response experiments using another cyclodextrin, γ -cyclodextrin. γ -cyclodextrin is less effective at sequestering cholesterol and as shown through Figure 5 and Table S6 in the revised manuscript had significantly less effect on glycine concentration dependent currents than M β CD. If the concentrations/temperatures used reduced the specificity to sequester other membrane lipids we would expect that M β CD and γ -cyclodextrin would produce similar results.

Published M β CD incubation conditions for SLC6 transporter characterisation:

Source	DOI	[Cyclodextrin] Range	Incubation	System
Laursen et al. (2018)	10.1074/jbc.M117.809046	3.8 – 15 mM	37°C for 30 min	HEK-293-MSR membranes
Scanlon (2001)	10.1021/bi010730z	3.8 – 7.5 mM	RT for 30 min	HEK-293 cells
Magnani et al. (2004)	10.1074/jbc.M400831200	0.96 – 20 mM	37°C for 30 min	T-Rex-SERT cells
Foster et al. (2008)	10.1111/j.1471-4159.2008.05262.x	5 mM	37°C for 30 min	LLC-PK ₁ cells
Nunez et al. (2008)	10.1111/j.1471-4159.2008.05292.x	10 – 50 mM	37°C for 1hr	Isolated membrane rafts
Liu et al. (2009)	10.1016/j.bbrc.2009.05.014	1 mM	RT for 45 min	CHO-K1 cells + X. laevis oocytes
Hong et al. (2010)	10.1074/jbc.M110.150565	2 – 10 mM	37°C for 15 min	HEK-293 cells + Rat synaptosomes
Jones et al., (2012)	10.1111/jnc.12007	2.5 – 5 mM	37°C for 30 min	HEK-293 cells

Published M β CD incubation conditions for *Xenopus laevis* oocytes:

Source	DOI	[Cyclodextrin] Range	Incubation
Sadler et al., (2004)	10.1095/biolreprod.103.026161	5 – 50 mM	RT for 15 min – 4 hours
Sadler et al., (2008)	10.1016/j.ydbio.2008.07.031	50 mM	RT for 3 – 24 hours
Santiago et al., (2001)	10.1074/jbc.M104563200	50 μ M	RT for 45 min
Cui et al., (2021)	10.3389/fphys.2021.652513	20 mM	RT for 3 hours

Comment:

Do authors know how much membrane cholesterol is removed by the treatment?

Reply:

To address this, we lysed control/M β CD treated oocytes and quantified the cholesterol levels in the supernatant using the AmplexTM Red Cholesterol Assay Kit. This assay demonstrated that M β CD treated oocytes contain 44% less cholesterol than the untreated controls (Figure 4 and lines 237-240 in revised manuscript). This result also speaks to the selectivity of our conditions as this reduction is similar to the level of cholesterol depletion reported in oocytes using other incubation conditions (Line 240).

Comment:

Re-complementation/replenishing of cholesterol

Reply:

We have addressed why re-complementation of cholesterol into oocytes was not possible in this instance in Lines 241-242 of the revised manuscript. Given the time constraints required for measuring transporter function before and after M β CD treatment, it was not possible to replenish cholesterol levels by subsequent M β CD-cholesterol treatment. In addition, previous work manipulating the cholesterol content of *Xenopus laevis* oocytes found this method of enriching cholesterol content to be of low reproducibility and efficiency (Lines 243-244). Therefore, as previously highlighted in this letter, we have incorporated additional experiments into the manuscript to provide stronger evidence this is a cholesterol mediated effect. Specifically, using γ -cyclodextrin and quantification of oocyte cholesterol content we have demonstrated the specificity of our methodology for cholesterol depletion.

Reviewer #1 Specific Comments

Comment:

It is not clear from Legends and Methods if oocytes are pre-incubated in the presence of CD for 30 min, and then further incubated in the presence of glycine (Fig 4) or bioactive lipids (Fig 5), or not. In the latter case what are the incubation conditions (temperature, time)?

Reply:

The methods, legends, and sections of the text pertaining to “glycine concentration responses” have been revised to clarify that baseline recordings were obtained from individual oocytes before they were incubated in cyclodextrin and that following treatment they were washed for 10 minutes in recording buffer before repeating recordings on the same cell (Lines 231-237, 646-649, 918-924). These sections have also been adapted to clarify how glycine concentrations were applied individually at room temperature until a stable current was achieved before removing glycine via washing and proceeding to the next concentration (Lines 631-633, 918-924).

Lines 682-687 have been adapted to highlight how bioactive lipids were applied following incubation with CD. Cells were washed for 10 minutes in recording buffer to remove residual CD and the experiments carried out as described earlier in the methods (Lines 655-657). Figure legends pertaining to CD inhibitor concentration responses have been updated to reflect this and to describe that concentrations were applied cumulatively and progressed once a stable plateau was achieved following administration of the previous concentration (Lines 974-976).

Comment:

Fig 9 does not seem to be mentioned in the text.

Reply:

Figure 9 from the original manuscript has been removed and its data incorporated into the text (Lines 329-330) as well as Figure 11 and Table S11 in the revised manuscript.

Reviewer #2 Specific Comments

Comment:

Manuscript is poorly written, including abstract.

Reply:

Sections of the manuscript have been re-written, including the abstract to improve overall quality and clarity.

Comment:

The effect of MCD on inhibitors are relatively minor, which would be difficult to tease out. Data is not at all well explained or interpreted. Why no MCD effect after 3 uM inhibitors?

Reply:

Our results demonstrate that M β CD treatment caused > 2-fold reduction in potency and up to a doubling of the reversibility of some of the inhibitors. Thus, we are unsure how they are considered minor.

We are also unsure what is meant by the comment “Why no MCD effect after 3uM inhibitors?” Therefore, we have addressed both possible interpretations in the revised manuscript. Firstly, if the comment is questioning why we did not test concentrations > 3 μ M, we have added an explanation in the methods (Lines 661-662) and Lines 344-346. These state that due to potential formation of lipid micelles affecting the amount of free inhibitor in solution, inhibitor concentrations did not exceed 3 μ M to ensure the inhibitors were always free in solution. Alternatively, if the comment is questioning why there is no difference in inhibition at 3 μ M between control and M β CD treated cells, we have added an interpretation in Lines 264-265 to say that combined these results show cholesterol depletion influences inhibitor potency but not maximal levels of inhibition.

Comment:

It is not clear washout experiments mean. Kd of inhibitors with or without MCD? How would this relate to cholesterol-inhibitor interaction? The data from 4 inhibitors seems not to be coherent from Fig 4 to Fig 6.

Reply:

To address this comment, we have added a sentence at the beginning of the washout experiment section to give them greater context (Lines 270-271). Briefly, the close association between the inhibitors and cholesterol in the MD simulations suggests that membrane cholesterol may play a role in the dissociation of the lipids from their binding site. Therefore, we used the washout experiments with or without M β CD to examine if membrane cholesterol influences the ability of the inhibitors to dissociate from their binding site. We are unsure what the comment “The data from 4 inhibitors seems not to be coherent from Fig 4 to Fig 6” means. Figure 4 (Figure 5 in the revised manuscript) corresponds to the effect of cyclodextrin treatment on glycine transport kinetics. This set of experiments assessed glycine transporter functionality, not its sensitivity to inhibitors, hence their absence. Figures 5 (Figure 6 in revised manuscript) and 6 (Figure 7 in revised manuscript) correspond to the concentration response and washout time course of the inhibitors tested, respectively. The only difference we can see between these two figures that may correspond to this comment is a spelling mistake in the names of the compounds, which has been amended in the revised manuscript.

Comment:

It is also unclear the purpose of the mutagenesis study. If they are aimed to illustrate the location where inhibitors interact with cholesterol, should cholesterol depletions be tested there with inhibitors?

Reply:

To address the first part of the comment we have added a section at the beginning of the mutagenesis section (Lines 291-298). Briefly, cholesterol is known to interact with other members of the SLC6 family (e.g. SERT). Mutagenesis of SERT residues that match the cholesterol site identified in our MD simulations disrupt the interaction with cholesterol and produces functional effects. These functional effects mimic the effect of cholesterol depletion via M β CD on SERT. Therefore, our mutagenesis section aimed to similarly disrupt cholesterol binding and examine if that produced the same functional effects as M β CD on GlyT2. As the aim of this section is to replicate the effect of M β CD using mutagenesis, it is not appropriate to deplete membrane cholesterol from oocytes expressing mutant transporters. This would result in cholesterol being removed, making it

impossible to determine if the effects observed were a result of perturbed cholesterol binding induced by the mutants, or due to the absence of cholesterol from the membrane.

November 25, 2022

Re: Life Science Alliance manuscript #LSA-2022-01708-TR

Prof. Megan L O'Mara
University of Queensland
Australian Institute for Bioengineering and Nanotechnology
The University of Queensland
St Lucia, Qld 4072
Australia

Dear Dr. O'Mara,

Thank you for submitting your revised manuscript entitled "Membrane cholesterol regulates inhibition and substrate transport by the glycine transporter, GlyT2" to Life Science Alliance. The manuscript has been seen by the original reviewers whose comments are appended below. While Reviewer 1 continues to be overall positive and satisfied with the revisions performed, Reviewer 2 instead claims that some important issues still remain that require to be addressed before the paper can be considered for publication at Life Science Alliance.

Our general policy is that papers are considered through only one revision cycle; however, given that the suggested control experiment is relatively important, we are open to one additional short round of revision. Please note that I will expect to make a final decision without additional reviewer input upon resubmission.

Please submit the final revision within two months, along with a letter that includes a point by point response to the remaining reviewer comments.

To upload the revised version of your manuscript, please log in to your account: <https://lsa.msubmit.net/cgi-bin/main.plex>
You will be guided to complete the submission of your revised manuscript and to fill in all necessary information.

B. MANUSCRIPT ORGANIZATION AND FORMATTING:

Sincerely,

Reviewer #1 (Comments to the Authors (Required)):

The authors have addressed most of my queries and comments in a satisfactory manner, and, as a result, the paper has improved.

Reviewer #2 (Comments to the Authors (Required)):

I find the rebuttal from authors on the main question of MCD specificity for cholesterol removal is very far from sufficient.

First of all, this is the only experimental evidence linking the whole project to cholesterol. It needs to prove that the effects they have observed are indeed due to the removal of cholesterol, not phospholipids. Without successful reversal in their assays by cholesterol repletion, it is not valid to conclude the involvement of cholesterol at all here. As I mentioned in previous review, I could not judge simulation results. Experimentally, current data does not support their conclusion.

Secondly, what others did or didn't do is irrelevant here.

Thirdly, experiment to replete cholesterol could be too difficult to do as authors claimed. Still, without this control experiment, one cannot claim cholesterol effects. They are MCD effects now. It should be noted that MCD removes cholesterol extremely fast, within first few minutes' exposure to cholesterol-containing membrane. How much MCD takes depends on cholesterol, or phospholipid, concentration gradient between the membrane and MCD. MCD takes cholesterol or phospholipid into its cavity rapidly and done with it, having reached an equilibrium or steady state. By the same token, repletion is as fast. Yes, it would take careful calibrations here as how much MCD/cholesterol complex to replete. However, it is absolutely critical and necessary to have this control experiment.

Without this control, other revisions are irrelevant.

Our responses to the Reviewer comments are in blue text.

Reviewer #1 (Comments to the Authors (Required)):

The authors have addressed most of my queries and comments in a satisfactory manner, and, as a result, the paper has improved.

We thank the Reviewer for their positive comments and support of our work.

Reviewer #2 Specific Comments

As the majority of Reviewer #2s comments revolve around the specificity of M β CD and their insistence on rescuing the effect with a M β CD-cholesterol complex we have incorporated our responses into one succinct reply. Where specific comments are being rebutted those sections have been bolded to orient the reader.

Reviewer #2 Comments:

This is the only experimental evidence linking the whole project to cholesterol. It needs to prove that effects they have observed are indeed due to the removal of cholesterol, not phospholipids. Without successful reversal in their assays by cholesterol repletion, it is not valid to conclude the involvement of cholesterol at all here. As I mentioned in previous review, I could not judge the simulation results. Experimentally, current data does not support their conclusion.

What others did or didn't do is irrelevant here.

Experiment to replete cholesterol could be too difficult to do as authors claimed. Still without this control experiment, one cannot claim cholesterol effects. They are MCD effects now. It should be noted that MCD removes cholesterol extremely fast, within first few minutes' exposure to cholesterol-containing membrane. How much MCD takes depends on cholesterol, or phospholipid, concentration gradient between the membrane and MCD. MCD takes cholesterol or phospholipid into its cavity rapidly and done with it, having reached an equilibrium or steady state. By the same token, repletion is as fast. Yes, it would take careful calibrations here as how much MCD/cholesterol complex to replete. However, it is absolutely critical and necessary to have this control experiment. Without this control, other revisions are irrelevant.

With all due respect, we reject the suggestion that the M β CD work is the **only experimental evidence linking the whole project to cholesterol**. Our paper consists of three experimentally diverse approaches: molecular dynamics simulations, cyclodextrin treatment and site-directed mutagenesis. Each of these approaches complement each other to support the role of membrane cholesterol in modulating the transport activity and pharmacological sensitivity of GlyT2. The molecular dynamics simulations, which Reviewer #2 says they are unable to judge, provide the first link to cholesterol. In the initial stages of our simulations, cholesterol spontaneously and reproducibly associates with a site formed by TMs 1a, 5 and 7. As described in the introduction, this site is believed to be conserved across neurotransmitter transporters in the SLC6 family and exert a modulatory effect on their activity. The site identified in our simulations is consistent with what has been identified in simulations of other family members and the initial pose of cholesterol is the same

as the orientation of cholesterol bound in this site in atomic structures of dDAT. Furthermore, as the simulations progress, they show the cholesterol molecule flipping in its orientation before inserting deeper into GlyT2 and interacting with our bioactive lipid inhibitors. Therefore, these simulations provide a foundation that implicates membrane cholesterol in the functional and pharmacological modulation of GlyT2.

To functionally validate this, we first used cyclodextrin treatment to primarily deplete membranes of cholesterol. In our previous rebuttal letter, we provided a table that displayed **what others have or haven't done** when using cyclodextrins in oocytes or to study SLC6 transporters. We believe this is particularly relevant as it establishes a precedent for conclusions that can be acceptably drawn from the types of data we have presented. This showed that the concentration/incubation times of our study were within the range of what has been previously used to investigate cholesterol effects and that replenishment of membrane cholesterol was not a prerequisite for concluding a specific cholesterol effect when using cyclodextrins.

As highlighted in our previous rebuttal letter, enrichment of oocyte membranes with cholesterol using a **MBCD-cholesterol complex** has been found to be lowly reproducible and inefficient. However, as Reviewer #2 deemed this a **critical control** we attempted these experiments. We similarly found this method to be lowly reproducible and inefficient. Although we observed some rescue of the effect, the time scale required made it difficult to distinguish the contribution of re-supplemented cholesterol versus changes in transporter expression over time (Lines 241-245 in second revised manuscript). Therefore, in our experimental set up this is not a suitable control. The original alternative approaches we undertook in determining the cholesterol content of oocytes and replicating the effect on transport kinetics using γ -cyclodextrin remains the most reliable way of supporting a specific cholesterol effect.

However, to incorporate Reviewer #2's concerns regarding **other phospholipids** we have altered our interpretation to highlight the results are consistent with a majority cholesterol effect (Line 256-258 in second revised manuscript), as M β CD is primarily used to deplete membrane cholesterol. Additionally, we have renamed that section of the results to "M β CD treatment alters the transport activity of GlyT2 and the potency and reversibility of some GlyT2 inhibitors." Finally, we used mutagenesis of cholesterol coordinating residues identified in the molecular dynamics simulations to show that destabilising mutations replicated the effect of cyclodextrin treatment. Thus, in this study we have used three distinct approaches that provide results that are consistent in supporting a role for membrane cholesterol in influencing the functionality and pharmacological sensitivity of GlyT2.

January 11, 2023

RE: Life Science Alliance Manuscript #LSA-2022-01708-TRR

Prof. Megan L O'Mara
University of Queensland
Australian Institute for Bioengineering and Nanotechnology
The University of Queensland
St Lucia, Qld 4072
Australia

Dear Dr. O'Mara,

Thank you for submitting your Research Article entitled "Membrane cholesterol regulates inhibition and substrate transport by the glycine transporter, GlyT2". It is a pleasure to let you know that your manuscript is now accepted for publication in Life Science Alliance. Congratulations on this interesting work.

DISTRIBUTION OF MATERIALS:

Again, congratulations on a very nice paper. I hope you found the review process to be constructive and are pleased with how the manuscript was handled editorially. We look forward to future exciting submissions from your lab.

Sincerely,
